# Contribution of the world's main dust source regions to the global cycle of desert dust

Jasper F. Kok[1], Adeyemi A. Adebiyi[1], Samuel Albani[2,3], Yves Balkanski[3], Ramiro Checa-Garcia[3], Mian Chin[4], Peter R. Colarco[4], Douglas S. Hamilton[5], Yue Huang[1], Akinori Ito[6], Martina Klose[7,§], Longlei Li[5], Natalie M. Mahowald[5], Ron L. Miller[8], Vincenzo Obiso[7,8], Carlos Pérez García-Pando[7,9], Adriana Rocha-Lima[10,11], and Jessica S. Wan[5,*]

[1]Department of Atmospheric and Oceanic Sciences, University of California, Los Angeles, CA 90095, USA
[2]Department of Environmental and Earth Sciences, University of Milano-Bicocca, Milano, Italy
[3]Laboratoire des Sciences du Climat et de l'Environnement, CEA-CNRS-UVSQ-UPSaclay, Gif-sur-Yvette, France
[4]Atmospheric Chemistry and Dynamics Laboratory, NASA Goddard Space Flight Center, Greenbelt, MD 20771, USA
[5]Department of Earth and Atmospheric Sciences, Cornell University, Ithaca, NY 14850, USA
[6]Yokohama Institute for Earth Sciences, JAMSTEC, Yokohama, Kanagawa 236-0001, Japan
[7]Barcelona Supercomputing Center (BSC), 08034 Barcelona, Spain
[8]NASA Goddard Institute for Space Studies, New York NY10025 USA
[9]ICREA, Catalan Institution for Research and Advanced Studies, 08010 Barcelona, Spain
[10]Physics Department, UMBC, Baltimore, Maryland, USA
[11]Joint Center Joint Center for Earth Systems Technology, UMBC, Baltimore, Maryland, USA
§Present address: Institute of Meteorology and Climate Research (IMK-TRO), Department Troposphere Research, Karlsruhe Institute of Technology (KIT), Karlsruhe, Germany
*Present address: Scripps Institution of Oceanography, University of California San Diego, La Jolla, CA, USA

*Correspondence to:* Jasper F. Kok (jfkok@ucla.edu)

**Abstract.** Even though desert dust is the most abundant aerosol by mass in Earth's atmosphere, the relative contributions of the world's major dust source regions to the global dust cycle remain poorly constrained. This problem hinders accounting for the potentially large impact of regional differences in dust properties on clouds, the Earth's energy balance, and terrestrial and marine biogeochemical cycles. Here, we constrain the contribution of each of the world's main dust source regions to the global dust cycle. We use an analytical framework that integrates an ensemble of global aerosol model simulations with observationally informed constraints on the dust size distribution, extinction efficiency, and regional dust aerosol optical depth (DAOD). We obtain a data set that constrains the relative contribution of each of nine major source regions to size-resolved dust emission, atmospheric loading, DAOD, concentration, and deposition flux. We find that the 22-29 Tg (one standard error range) global loading of dust with geometric diameter up to 20 μm is partitioned as follows: North African source regions contribute ~50% (11-15 Tg), Asian source regions contribute ~40% (8-13 Tg), and North American and Southern Hemisphere regions contribute ~10% (1.8-3.2 Tg). Current models might on average be overestimating the contribution of North African sources to atmospheric dust loading at ~65%, while underestimating the contribution of Asian dust at ~30%.. Our results further show that each source region's dust loading peaks in local spring and summer, which is partially driven by increased

dust lifetime in those seasons. We also quantify the dust deposition flux to the Amazon rainforest to be ~10 Tg/year, which is a factor of 2-3 less than inferred from satellite data by previous work that likely overestimated dust deposition by underestimating the dust mass extinction efficiency. The data obtained in this paper can be used to obtain improved constraints on dust impacts on clouds, climate, biogeochemical cycles, and other parts of the Earth system.

## 1 Introduction

Desert dust is likely the most abundant aerosol type by mass (Kinne et al., 2006; Kok et al., 2017) and produces a range of important impacts on the Earth system, including on clouds, the Earth's energy and water budgets, and biogeochemical cycles (Shao et al., 2011; Mahowald et al., 2014; Miller et al., 2014). These impacts are spatially heterogeneous, not only because dust loading itself varies substantially between regions, but also because the properties of dust depend on mineralogy, which varies with the region of origin (Claquin et al., 1999; Grousset and Biscaye, 2005; Journet et al., 2014). Consequently, determining dust impacts on the Earth system requires not only constraints on global dust loading (Huneeus et al., 2011; Kok et al., 2017), but also on the emission, loading, and deposition generated by individual source regions. However, estimates of the contributions of the world's major source regions to the global dust cycle diverge widely. For instance, model estimates of emissions from the main source regions vary by up to an order of magnitude between different global aerosol model simulations (Huneeus et al., 2011; Wu et al., 2020).

There are several reasons why this poor understanding of the contribution of each source region to the global dust cycle hinders quantification of dust impacts on the Earth system. First, since dust loading is spatially heterogeneous, constraining regional dust loading is a prerequisite to constraining dust impacts on regional climate, weather, air quality, and the hydrological cycle (Seinfeld et al., 2004; Engelstaedter et al., 2006; Huang et al., 2014; Vinoj et al., 2014; Sharma and Miller, 2017; Kok et al., 2018). Second, dust deposition records indicate that atmospheric dust loading has been highly variable, both between glacial and interglacial periods and from pre-industrial to modern times (Petit et al., 1999; McConnell et al., 2007; Albani et al., 2018; Hooper and Marx, 2018). Inferring the impacts of these large changes in dust loading on the Earth system requires knowledge of the origin of dust deposited to each measurement site (Mahowald et al., 2010). Finally, many dust impacts on the Earth system are sensitive to dust mineralogy, which varies both within and between major source regions (Biscaye, 1965; Claquin et al., 1999; Di Biagio et al., 2017). Examples of dust impacts that are sensitive to mineralogy include dust direct radiative effects (Balkanski et al., 2007; Perlwitz et al., 2015b; Scanza et al., 2015; Di Biagio et al., 2019), dust interactions with clouds through dust serving as ice nucleating particles (Atkinson et al., 2013; Shi and Liu, 2019), and dust impacts on biogeochemistry through the deposition of micronutrients such as iron and phosphorus (Swap et al., 1992; Jickells et al., 2005; Journet et al., 2008; Schroth et al., 2009). Since dust has potentially doubled since pre-industrial times (Mahowald et al., 2010; Hooper and Marx, 2018), some of these effects might have produced a marine biogeochemical response (Hamilton et al., 2020; Ito et al.,

2020) that might have resulted in a substantial global indirect radiative forcing (Mahowald, 2011). As such, constraining source-specific dust emissions and loading is critical to constraining global climate sensitivity (Andreae et al., 2005; Kiehl, 2007).

Many past studies of the contributions of the main dust source regions to the global dust cycle have been based on global aerosol model simulations (Tanaka and Chiba, 2006; Chin et al., 2007; Huneeus et al., 2011; Wu et al., 2020). These simulations exhibit substantial biases when compared to observations of dust abundance and of dust microphysical properties like size distribution and mass extinction efficiency (Kok et al., 2014a; Ansmann et al., 2017; Adebiyi et al., 2020; Checa-Garcia et al., 2020). Recognizing this problem, Ginoux et al. (2012) made an important advance towards more accurate constraints on the regional contributions to the global dust cycle by using extensive dust aerosol optical depth data from the Moderate Resolution Imaging Spectroradiometer (MODIS) Deep Blue algorithm to obtain a fine-grained map of the frequency of activation of dust sources. They then used this result to scale emissions in a global aerosol model simulation tuned to a specific global emission rate (1223 Tg/year for dust up to 12 μm diameter) obtained in a previous modeling study (Ginoux et al., 2001), thereby obtaining results for the emission rate generated by each main source region. Another recent study by Albani et al. (2014) regionally tuned model sources to match concentration, deposition, and aerosol optical depth observations using optimal estimation combined with expert opinion to incorporate geochemical tracer information in dust deposition measurements (Albani et al., 2014). Although both these studies are steps towards obtaining more reliable constraints on the contribution of each main source region to the global dust cycle, remaining deficiencies include (1) substantial biases in dust microphysical properties such as size distribution and extinction efficiency (Adebiyi and Kok, 2020; Adebiyi et al., 2020), (2) the use of a single model to represent dust transport and emission even though the spread in predictions between models is large (Huneeus et al., 2011; Checa-Garcia et al., 2020; Wu et al., 2020), and (3) the lack of robust uncertainty estimates in the contributions of the different source regions that can be propagated into calculated dust impacts such as dust radiative forcing. As such, although clear progress has been made in understanding the relative contributions of the world's source regions to the global dust cycle, current knowledge is not yet sufficient to constrain regionally varying dust impacts or to reliably inform the provenance of dust in deposition records.

Here we obtain improved constraints on the contributions from the world's main dust source regions to the global cycle of desert dust. We do so in Section 2 by building on the improved representation of the global dust cycle that we obtained in a companion paper (Kok et al., 2021) by integrating an ensemble of global aerosol model simulations with observational constraints on the properties and abundance of atmospheric dust. We obtain a data set that constrains the contributions of each of nine major source regions to size-resolved dust emission, loading, dust aerosol optical depth (DAOD), concentration, and deposition. Our results in Section 3 suggest that most models overestimate the contribution of African dust to the global dust cycle, while they underestimate the contribution of Asian dust. As discussed in Section 4, our data set can be used both to

100 improve global dust cycle simulations and to constrain dust impacts on the Earth system, including on regional and global climate, weather, air quality, ecosystems, and the hydrological cycle.

## 2 Methods

We seek to constrain the contributions of the world's major source regions to the global dust cycle. We do so by analyzing constraints obtained in our companion paper (Kok et al., 2021) on the dust loading, concentration, emission flux, wet and dry
deposition fluxes, and DAOD generated by each source region. These constraints were obtained through an analytical framework that used an inverse model to integrate an ensemble of model simulations with observational constraints on dust properties and abundance. Briefly, we used simulations from six different global models of dust emitted by each of nine major source regions (Fig. 1) and for each model particle bin (or mode) extending to a maximum diameter of 20 µm. These emissions implicitly include both natural and anthropogenic dust and together account for over 99% of global low latitude desert dust
emissions (our methodology excludes high latitude dust). We then used optimal estimation to determine how many units of dust loading from each source region and particle bin produces best agreement against observational constraints on the dust size distribution, extinction efficiency, and regional dust aerosol optical depth. This approach yielded constraints on the dust loading, optical depth, concentration, and emission and deposition fluxes that are resolved by location, season, and particle size. These constraints include uncertainties propagated from the spread in model simulation results and in the observationally
informed constraints on dust microphysical properties and regional DAOD. The companion paper (Kok et al., 2021) shows that this approach resulted in a representation of the Northern Hemisphere global dust cycle that is substantially more accurate than obtained from both a large number of climate and chemical transport model simulations and the MERRA-2 dust product (Buchard et al., 2017), with modest improvements for the Southern Hemisphere.

Below, we first describe how we used the results from this inverse model to obtain the fractional contribution to dust loading, concentration, emission flux, wet and dry deposition fluxes, and DAOD generated by each source region (Section 2.1). For the remainder of this paper, we refer to these results as the "inverse model" results. We then describe our analysis of the dust emission, dust loading, and DAOD generated by each source region in the ensemble of AeroCom Phase 1 global aerosol model simulations (Huneeus et al., 2011) (Section 2.2).

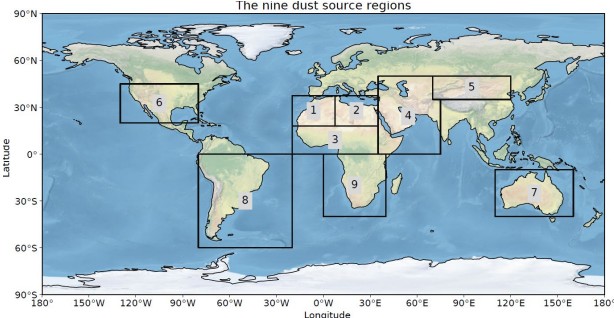

**Figure 1. Coordinates of the nine main source regions used in this study.** The nine source regions are: (1) western North Africa, (2) eastern North Africa, (3) Southern Sahara & Sahel, (4) Middle East & Central Asia (which includes the Horn of Africa), (5) East Asia, (6) North America, (7) Australia, (8) South America, and (9) Southern Africa. Exact coordinates for these regions are given in Kok et al. (2021). After Kok et al. (2021); made with Natural Earth.

## 2.1 Attribution of the global dust cycle to the different dust source regions

The analysis in Kok et al. (2021) integrated an ensemble of global aerosol model simulations with observational constraints on dust properties and abundance to obtain the size-resolved dust optical depth, column loading, emission flux, deposition flux, and concentration for each season and source region. Here we used these inverse modeling results to constrain the fractional contribution of each source region to DAOD ($\check{f}_{\tau_{r,s}}$), column loading ($\check{f}_{l_{r,s}}$), dust concentration ($\check{f}_{C_{r,s}}$), and dust deposition flux ($\check{f}_{D_{r,s}}$). These fields are a function of longitude ($\theta$), latitude ($\phi$), pressure level ($P$; in the case of dust concentration), and season (subscript $s$). That is,

$$\check{f}_{\tau_{r,s}} = \check{\tau}_{r,s} / \sum_{r=1}^{N_{\text{sreg}}} \check{\tau}_{r,s}, \tag{1}$$

$$\check{f}_{l_{r,s}} = \check{l}_{r,s} / \sum_{r=1}^{N_{\text{sreg}}} \check{l}_{r,s}, \tag{2}$$

$$\check{f}_{C_{r,s}} = \check{C}_{r,s} / \sum_{r=1}^{N_{\text{sreg}}} \check{C}_{r,s}, \tag{3}$$

$$\check{f}_{D_{r,s}} = \check{D}_{r,s} / \sum_{r=1}^{N_{\text{sreg}}} \check{D}_{r,s}, \tag{4}$$

where $\check{\tau}_{r,s}$, $\check{l}_{r,s}$, $\check{C}_{r,s}$, and $\check{D}_{r,s}$ are respectively the spatially-resolved bulk DAOD, dust loading, concentration, and total (wet and dry) deposition flux generated by dust from source season $r$ in season $s$, obtained from the analysis in Kok et al. (2021). These four products account for bulk dust with a geometric diameter up to 20 μm (PM$_{20}$). We also obtained the corresponding size-resolved fields for different particle size bins $k$, namely for $0.2 - 0.5$, $0.5 - 1.0$, $1.0 - 2.5$, $2.5 - 5.0$, $5.0 - 10$, and $10 - 20$ μm (see the Supplement for details).

The DAOD, column loading, concentration, and deposition fields used in Eqs. (1)-(4) are probability distributions that account for the propagation of errors in the observational constraints and modeling results that were used as inputs to generate these fields (see section 2.5 in Kok et al., 2021). The fractional contribution of each source region to DAOD, column loading, concentration, and deposition flux obtained by Eqs. -(4) are thus also probability distributions. We added the mean and median, the upper and lower one standard error estimates, and the upper and lower two standard error estimates to the Dust Constraints from joint Experimental-Modeling-Observational Analysis (DustCOMM) dataset (Adebiyi et al., 2020), which is available at https://dustcomm.atmos.ucla.edu/.

**2.3 Obtaining emission, loading, and DAOD per source region from AeroCom models**

As described in more detail in the companion paper (Kok et al., 2021), we analyzed 13 AeroCom Phase I simulations of the dust cycle in the year 2000 (Huneeus et al., 2011) for comparison against our inverse model's results of the contribution of each source region to the global dust cycle. Although newer global aerosol model ensembles are available, such as the
AeroCom phase III (Gliss et al., 2021) and CMIP5 model ensembles (Wu et al., 2020), only the dust component of AeroCom Phase I models has been analyzed in sufficient detail (Huneeus et al., 2011) for comparison against the results of our study. However, the error of newer model ensembles relative to various measurements appears to be similar to those for AeroCom Phase I models (see further discussion in Kok et al., 2021) and emissions per source region of CMIP5 models are relatively similar to those of the AeroCom Phase I models analyzed here (see Table 4 in Wu et al., 2020).

Since AeroCom simulations did not track the source region of atmospheric dust after emission, we used our ensemble of model simulations (see Table 1 in Kok et al., 2021) to estimate regional differences in the conversion of source-specific dust emission to source-specific global loading, and the conversion of source-specific global loading to source-specific global DAOD. Specifically, we estimated the global loading (Tg) generated by dust emitted from source region $r$ as simulated by an AeroCom
model as:

$$\tilde{L}_r^{\text{Aer}} = \tilde{F}_r^{\text{Aer}}\tilde{T}_{\text{glob}}^{\text{Aer}}\frac{\breve{T}_r}{\breve{T}_{\text{glob}}},$$  (5)

where $\tilde{F}_r^{\text{Aer}}$ is the bulk emission flux generated by source region $r$ simulated by a given AeroCom model, $\tilde{T}_{\text{glob}}^{\text{Aer}}$ is the global bulk (mass-weighted) dust lifetime simulated by the AeroCom model (obtained from Table 3 in Huneeus et al. (2011)), and $\breve{T}_r$ and $\breve{T}_{\text{glob}}$ are respectively the mean bulk lifetimes for source region $r$ and for dust from all source regions (both obtained from our inverse model results, see Fig. 3a below). As such, $\breve{T}_r/\breve{T}_{\text{glob}}$ estimates the ratio of the lifetime of dust from source
region $r$ to the global dust lifetime. This ratio is used in Eq. (5) to correct the global dust lifetime simulated by an AeroCom model to the lifetime for dust emitted from source region $r$, which in turn is used to calculate the dust loading generated by source region $r$ from its emission flux.

We furthermore obtain the DAOD generated by the dust emitted from source region $r$ as simulated by an AeroCom model as:

$$\tilde{\tau}_r^{\text{Aer}} = \frac{\tilde{L}_r^{\text{Aer}}}{A_{\text{Earth}}}\tilde{\epsilon}_{\text{glob}}^{\text{Aer}}\frac{\breve{\epsilon}_r}{\breve{\epsilon}_{\text{glob}}},$$  (6)

where $A_{\text{Earth}}$ is the Earth's surface area (m$^2$) and $\tilde{\epsilon}_{\text{glob}}^{\text{Aer}}$ is the bulk dust mass extinction efficiency (MEE) simulated by a given
AeroCom model (obtained from Table 3 in Huneeus et al. (2011)). Similar to the approach used in Eq. (5), $\breve{\epsilon}_r$ and $\breve{\epsilon}_{\text{glob}}$ are respectively the bulk dust MEE for source region $r$ and for dust from all source regions (both also obtained from our inverse model results, see Table 3 in Kok et al. (2021)). Note that using $\breve{T}_r/\breve{T}_{\text{glob}}$ and $\breve{\epsilon}_r/\breve{\epsilon}_{\text{glob}}$ from our model ensemble to approximate the lifetime and MEE per source region in AeroCom models will introduce some error; however, because $\breve{T}_r/\breve{T}_{\text{glob}}$ and

$\breve{\epsilon}_r/\breve{\epsilon}_{\mathrm{glob}}$ are dimensionless ratios, we expect these errors to be relatively small compared to other errors. This is also indicated by the limited differences between the fractional contributions to emission, loading, and DAOD for the different source regions (Fig. 2).

## 3. Results

We report each source region's contributions to the global dust cycle on annual and seasonal timescales in Sections 3.1 and 3.2, respectively. We then report the spatial distribution of dust column loading and DAOD in Section 3.3 and the spatial distribution of dust deposition fluxes in Section 3.4.

### 3.1 Constraints on each source region's contribution to the global dust cycle

We obtained each source region's absolute and fractional contributions to the global dust emission and deposition flux, the global dust loading, and the global DAOD (see Table 1 and Fig. 2). For comparison, we also obtained these properties for our ensemble of simulations and for the AeroCom Phase I model ensemble (see Section 2.2).

Our inverse model results on each source region's contribution to the global dust cycle show some notable differences from our model ensemble and the AeroCom Phase I ensemble. First, the inverse model results indicate that $PM_{20}$ dust emission fluxes for all source regions are substantially greater than most models include (Table 1; also see Kok et al., 2021). This is in part because many models simulate dust up to a maximum geometric diameter smaller than 20 μm (see further discussion in Kok et al., 2021) and in part because most models that do simulate dust with diameter larger than 10 μm substantially underestimate dust emission and loading in the 10-20 μm diameter range (Adebiyi and Kok, 2020; Huang et al., 2021). Accounting for this additional coarse dust in the atmosphere is important because it produces a substantial direct radiative forcing (Ryder et al., 2019; Adebiyi and Kok, 2020) and because it accounts for a large fraction of dust deposition fluxes to marine and terrestrial ecosystems (see below).

A second important difference between our results and that of the two model ensembles is in the fractional contributions per source region to the global dust cycle. We find that the model ensemble used here better matches the fractional contribution obtained by the inverse model than does the AeroCom phase I model. This might be because of a closer match in the simulated time period (most simulations in our ensemble are for the 2004-2008 period for which inverse model results were obtained, whereas AeroCom simulations were for the year 2000) or because of improvements in parameterizations of dust emission and other dust processes. However, both multi-model ensemble means show that ~60-65% of dust loading and DAOD is generated by North African source regions, ~20-25% from the Middle East & Central Asia, ~5-9% from East Asia, <1% from North America, and ~5% from the Southern Hemisphere source regions (Figs. 2b). In contrast, we find that North African dust

contributes only about half of the global dust loading (one standard error range of 11-15 Tg) and DAOD (0.013-0.015). Specifically, we find that the Southern Sahara & Sahel source region contributes about ~15% of global dust loading (with a large one standard error range of 1.6-5.4 Tg), and that western North Africa (~20%; 4.2-7.1 Tg) likely contributes substantially more dust than Eastern North Africa (~15%; 2.2-6.0 Tg). These fractional contributions of western and eastern North Africa are substantially less than the average for models in our ensemble and in the AeroCom ensemble, both of which obtained a ~25-30% contribution for both regions. These findings that most models overestimate the fractional contribution of North Africa to global dust loading and that western North Africa generates a larger dust loading than eastern North Africa appear to be consistent with satellite observations (Engelstaedter et al., 2006; Shindell et al., 2013). We furthermore find that the ~15% contributions of the Southern Sahara & Sahel to global dust loading is similar to that simulated by models in our ensemble but that AeroCom models simulated a contribution of on average only ~6%, thereby possibly underestimating the contribution from the Southern Sahara & Sahel by about a factor of two. Our finding of a larger contribution to global dust loading from the Southern Sahara & Sahel is consistent with the fact that this source region includes the Bodélé Depression, which is a major dust source (Warren et al., 2007). Nonetheless, our results add to an emerging consensus (Glaser et al., 2015; Bozlaker et al., 2018; Yu et al., 2020) that dust from the Bodélé Depression accounts for much less than the previously-proposed ~half of North African dust transported across the Atlantic (Koren et al., 2006; Washington et al., 2009; Evan et al., 2015).

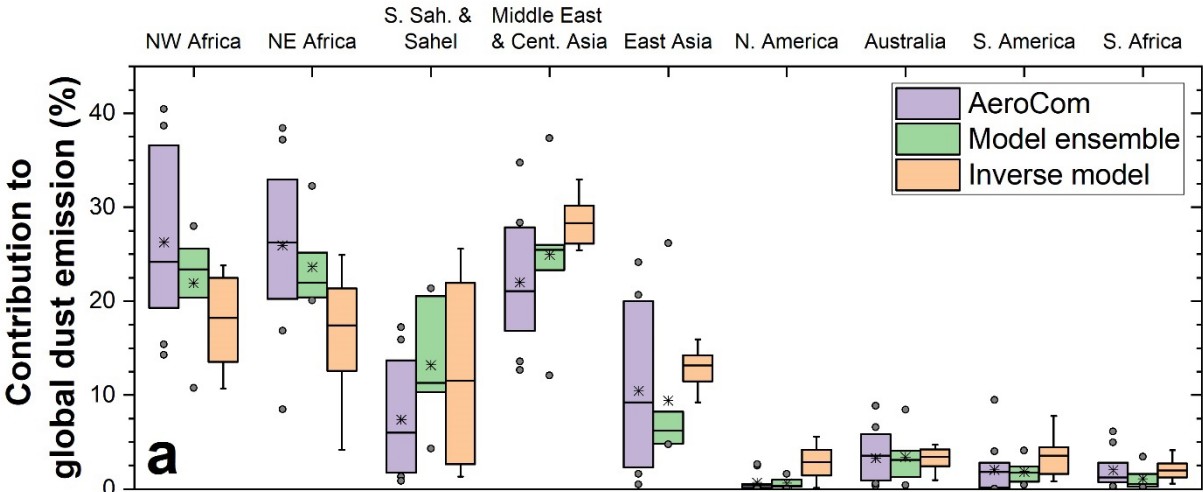

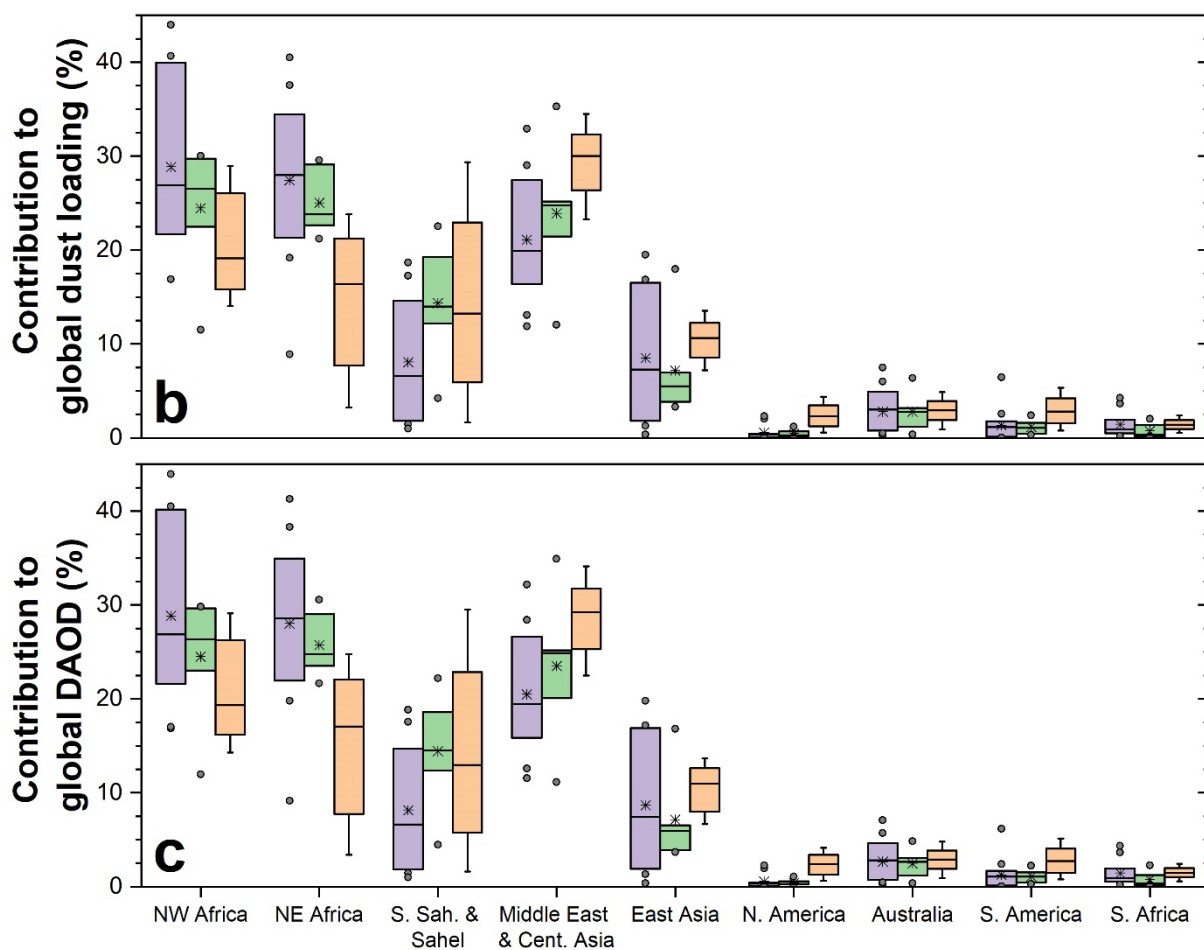

**Figure 2. Fractional contribution of each source region to the global dust cycle.** Shown are the fractional contributions to the annual global dust emission (and deposition) flux (**a**), the global dust loading (**b**), and the global DAOD (**c**) for the AeroCom ensemble (purple boxes; see Section 2.2) our model ensemble (green boxes) and for the inverse model (orange boxes). Box boundaries approximately denote the one standard error range (i.e., boxes contain 9 out of 13 AeroCom simulations, 4 out of 6 model ensemble members, and the 68% probability range for the inverse model's results), gray circles denote the individual simulation results outside of this range, whiskers denote the 95% confidence interval for the inverse model results, horizontal solid lines denote the median result, and stars denote the mean result. Results for each season are shown in Figs. S1-S4.

After the North African source regions, we find that Asian source regions account for the bulk of the remainder of global dust emissions, loading, and DAOD. In particular, the Middle Eastern and Central Asian source regions account for ~30% of global dust loading (6.1-9.4 Tg), which is more than the ~20-25% estimated from the AeroCom ensemble and our model ensemble. We further find that East Asian source regions account for another ~11% of global dust loading (2.0-3.4 Tg), which is more than the ~7% estimated from the two model ensembles. Overall, we find that Asian dust accounts for ~40% of global dust loading.

**Table 1. Contributions of the world's nine major source regions to the global dust cycle.** Listed are median values for the AeroCom Phase I ensemble and the inverse model's results, with one standard error intervals listed in parentheses. For the AeroCom ensemble, the one standard range was obtained from the range spanned by the 9 central model results out of the 13 total model results, which corresponds to the central 69% of model results. For the inverse model results, the one error range was obtained from the central 68% of results from a

 large number ($10^3$) bootstrap iterations (see Kok et al., 2021). Note that inverse model results are for dust with $D \leq 20$ µm, whereas the size range accounted for by AeroCom models differs for each model (see Huneeus et al. (2011)).

| Source region | Annual dust emission and deposition rate ($\times 10^3$ Tg/year) | | Percentage of annual dust emission and deposition | | Dust loading (Tg) | | Percentage of dust loading | | Dust AOD ($\times 10^3$) | | Percentage of dust AOD | | Mass extinction efficiency (m²/g) | |
|---|---|---|---|---|---|---|---|---|---|---|---|---|---|---|
| | Aero-Com | Inverse model | Aero-Com | Inverse model | Aero-Com | Inverse model | Aero-Com | Inverse model | Aero-Com | Inverse model | Aero-Com | Inverse model | Aero-Com | Inverse model |
| All source regions | 1.7 (1.2-3.1) | 4.7 (3.4-9.1) | 100 (99-100)* | 100 (99-100)* | 20 (14-23) | 26 (22-30) | 100 (99-100) | 100 (99-100) | 30 (21-35) | 27 (24-30) | 100 (99-100) | 100 (99-100) | 0.69 (0.60-0.96) | 0.54 (0.47-0.62) |
| All of North Africa | 1.0 (0.5-1.7) | 2.1 (1.6-4.3) | 60 (55-69) | 46 (44-49) | 12 (8-18) | 13 (11-15) | 66 (59-74) | 50 (47-55) | 20 (13-22) | 14 (13-15) | 66 (60-74) | 50 (48-55) | 0.69 (0.61-0.97) | 0.55 (0.48-0.63) |
| All of the Southern Hemisphere | 0.10 (0.03-0.20) | 0.47 (0.30-0.78) | 6 (2-11) | 9 (7-12) | 0.9 (0.3-1.5) | 1.8 (1.3-2.4) | 5 (1-8) | 7 (5-9) | 1 (0-2) | 2 (1-2) | 5 (1-8) | 7 (5-9) | 0.65 (0.57-0.93) | 0.53 (0.47-0.61) |
| Western North Africa | 0.41 (0.28-0.60) | 0.88 (0.64-1.44) | 24 (19-37) | 18 (14-22) | 5.0 (3.3-6.2) | 5.2 (3.8-6.9) | 27 (22-40) | 19 (16-26) | 7.4 (5.2-11.1) | 5.6 (4.2-7.1) | 27 (22-40) | 19 (16-26) | 0.68 (0.60-0.97) | 0.55 (0.48-0.63) |
| Eastern North Africa | 0.44 (0.27-0.82) | 0.72 (0.47-1.11) | 26 (20-33) | 16 (7-21) | 5.3 (3.8-8.2) | 4.3 (2.0-5.5) | 28 (21-34) | 16 (8-21) | 8.1 (6.0-10.5) | 4.8 (2.2-6.0) | 29 (22-35) | 17 (8-22) | 0.70 (0.62-0.99) | 0.56 (0.49-0.64) |
| Southern Sahara & Sahel | 0.10 (0.03-0.29) | 0.56 (0.15-1.77) | 6 (2-14) | 13 (4-20) | 1.4 (0.4-2.1) | 3.5 (1.5-5.6) | 7 (2-15) | 14 (6-22) | 2.1 (0.5-2.7) | 3.8 (1.6-5.4) | 7 (2-15) | 13 (6-22) | 0.69 (0.61-0.98) | 0.53 (0.46-0.61) |
| Middle East / Central Asia | 0.34 (0.19-0.56) | 1.38 (0.97-2.59) | 21 (17-28) | 29 (27-32) | 4.5 (1.8-5.7) | 7.7 (6.0-9.3) | 20 (16-27) | 30 (26-32) | 4.9 (4.0-8.0) | 8.0 (6.4-9.1) | 19 (16-27) | 29 (25-32) | 0.66 (0.59-0.94) | 0.52 (0.46-0.60) |
| East Asia | 0.18 (0.02-0.31) | 0.58 (0.42-1.12) | 9 (2-20) | 13 (10-15) | 1.8 (0.2-2.7) | 2.7 (2.0-3.4) | 7 (2-17) | 11 (9-12) | 2.1 (0.4-5.9) | 3.0 (2.0-3.6) | 7 (2-17) | 11 (8-13) | 0.70 (0.62-0.99) | 0.53 (0.46-0.63) |
| North America | 0.01 (0.00-0.01) | 0.13 (0.05-0.24) | 0 (0-1) | 3 (1-4) | 0.1 (0.0-0.1) | 0.6 (0.3-0.9) | 0 (0-0) | 2 (1-3) | 0.1 (0.0-0.1) | 0.6 (0.4-0.9) | 0 (0-0) | 2 (1-3) | 0.67 (0.59-0.94) | 0.54 (0.46-0.64) |
| Australia | 0.04 (0.01-0.09) | 0.16 (0.09-0.29) | 4 (1-6) | 3 (2-5) | 0.3 (0.1-0.9) | 0.8 (0.5-1.1) | 3 (1-5) | 3 (2-4) | 0.4 (0.2-1.1) | 0.8 (0.5-1.1) | 3 (1-5) | 3 (2-4) | 0.64 (0.57-0.91) | 0.53 (0.46-0.61) |
| South America | 0.02 (0.00-0.07) | 0.19 (0.10-0.35) | 2 (0-3) | 4 (2-6) | 0.1 (0.0-0.5) | 0.7 (0.4-1.1) | 1 (0-2) | 3 (2-4) | 0.1 (0.0-0.5) | 0.7 (0.4-1.1) | 1 (0-2) | 3 (2-4) | 0.65 (0.57-0.91) | 0.52 (0.45-0.60) |

| Southern Africa | 0.02 (0.01-0.05) | 0.10 (0.06-0.19) | 1 (1-3) | 2 (1-3) | 0.2 (0.1-0.4) | 0.4 (0.2-0.5) | 1 (1-2) | 1 (1-2) | 0.4 (0.1-0.5) | 0.4 (0.3-0.5) | 1 (1-2) | 1 (1-2) | 0.69 (0.61-0.97) | 0.57 (0.50-0.66) |

\*The fraction of global emissions of desert dust accounted for by the nine source regions is calculated by using simulated global emissions in the latitude band of 50S – 50N. This was done to exclude high latitude dust emissions, which the inverse model does not account for.

We find that the lesser source regions of North and South America, Australia, and South Africa account for ~10% of the global dust loading (1.8-3.2 Tg). This is substantially more than the ~5-6% of the global dust loading that both model ensembles estimate for these minor source regions. In particular, we find that the relative contribution of North America to global dust loading is ~2.5% (0.3-0.9 Tg), or ~5 times greater than estimated by both model ensembles (see further discussion in Section 3.4.1). Similarly, the relative contribution of South America is approximately twice as large (~3% of global dust loading (0.4-1.1 Tg). However, there are large uncertainties in our estimates for these minor source regions because it is difficult to obtain accurate constraints on the DAOD over those regions. In particular, the contributions of other aerosol species (e.g., sea spray) to total AOD can be larger than that due to dust (Ridley et al., 2016), which propagates into large uncertainties for our constraint on dust emissions, loading, and DAOD generated by these source regions (see Table 1). Nonetheless, our results further confirm that the global dust cycle is dominated by Northern Hemisphere dust, with Southern Hemisphere dust accounting for less than 10% of global dust loading.

The moderate differences between each source region's fractional contribution to global dust loading and emission/deposition fluxes (Fig. 2) are due to differences in the lifetimes of dust from each source region (Fig. 3). We find that atmospheric lifetimes are largest for dust emitted from the world's main source regions in North Africa and the Middle East & Central Asia. As discussed in more detail in Section 3.2, this likely occurs because dust emitted from these regions experiences strong convection, lofting the dust to greater heights in the atmosphere (Cakmur et al., 2004). The relatively large lifetime of North African dust causes it to account for half (~50%) of the global dust loading and DAOD, even though it accounts for somewhat less than half (~46%) of the global $PM_{20}$ dust emissions (Fig. 2 and Table 1). Differences in size-resolved lifetimes between different source regions (Figure S5) also drive small differences in the MEE between the different source regions (Table 1; note that we do not account for regional differences in MEE due to differences in dust mineralogy (Perlwitz et al., 2015a; Scanza et al., 2015)).

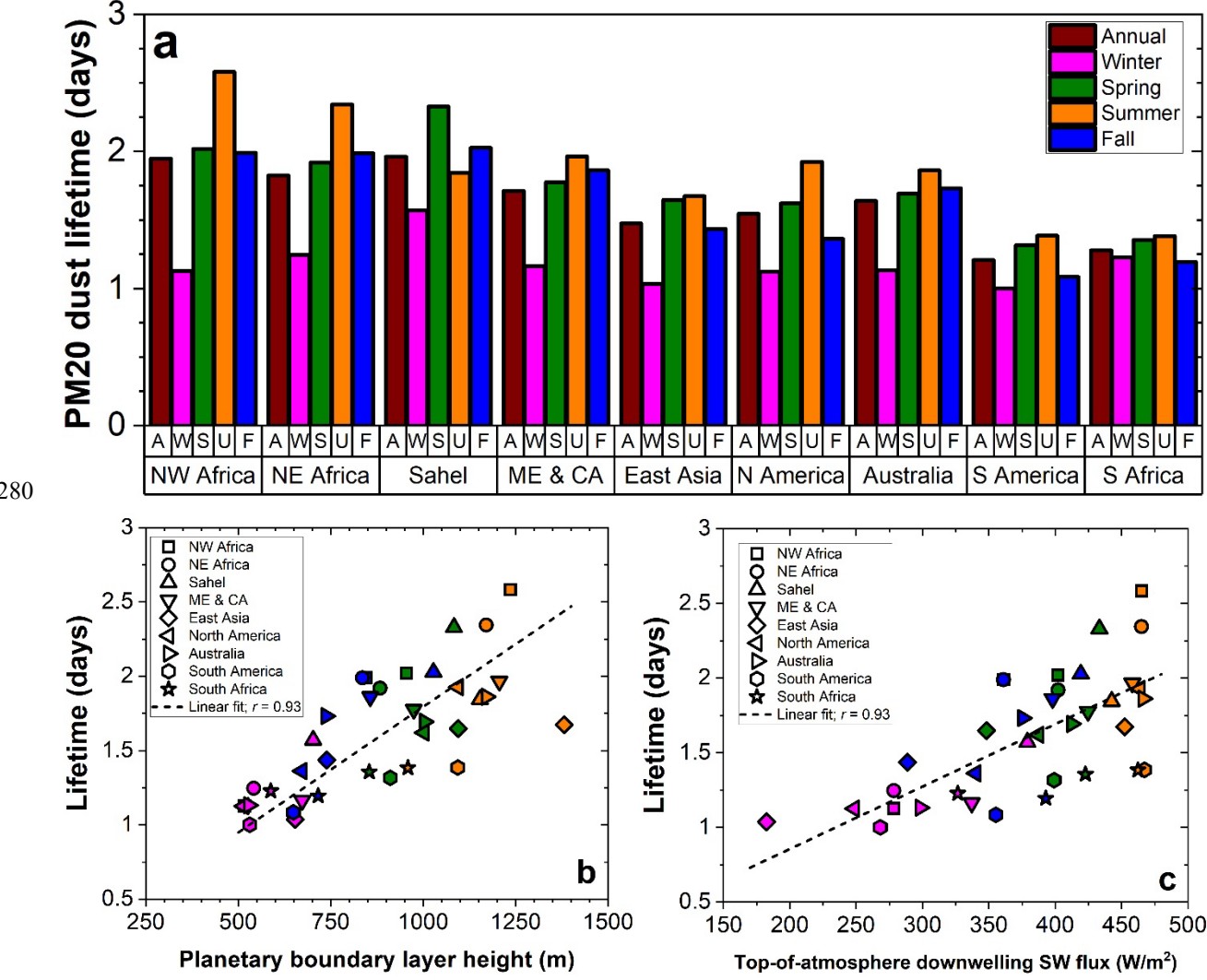

**Figure 3. Seasonal variation of dust lifetime for the different source regions.** (**a**) Mass-weighted lifetime of $PM_{20}$ dust for each source region on an annual basis (A; brown bars) and for each source region's local winter (W), spring (S), summer (U), and fall (F) seasons (magenta, green, orange, and blue bars). Variations in dust lifetime are largely explained by differences in downwelling SW flux (panel **b**) and planetary boundary layer (PBL) height (panel **c**). This indicates that the increased modeled dust lifetime in spring and summer is due to increased convection, which lofts dust to greater altitudes. Colors of symbols in panels **b** and **c** denote the season per the color scheme in panel **a**. The TOA downwelling SW flux was obtained from Wong and Chow (2001), the seasonally averaged PBL height was obtained from the GISS Model E2.1 simulation (Kelley et al., 2020), and the mass-weighted dust lifetimes were obtained from the ratio of the $PM_{20}$ dust loading with the deposition flux for each source region and season. Size-resolved annual and seasonal dust lifetimes of the individual model ensemble simulations used in the inverse model are shown in Figs. S5 – S9.

## 3.2 Seasonality of each source region's dust cycle

We further analyzed our results to obtain the seasonality of each source region's dust cycle (Fig. 4). We find that only ~15-30% of the total $PM_{20}$ deposition flux is due to wet deposition, with some variability with season and source region. This dominance of deposition fluxes by dry deposition in all seasons occurs because coarse dust dominates the total emission flux and those particles are predominantly removed through dry deposition (e.g., Miller et al., 2006). However, most of this dry deposition flux is due to coarse dust depositing close to source regions, and wet deposition remains dominant further from source regions (Yu et al., 2019; van der Does et al., 2020). All source regions show peaks in dust loading and DAOD in spring or summer, and these seasons are substantially dustier than the fall and winter seasons for both the Northern and the Southern Hemispheres. Our results are consistent with well-known features of the seasonality of the dust cycle, such as spring/summer-time peaks in Saharan dust loading, spring-time peaks in Sahelian and East Asian dust loading, and spring/summer-time peaks in Australian dust loading (Goudie and Middleton, 2001; Prospero et al., 2002; Ekstrom et al., 2004; Ginoux et al., 2012; Knippertz and Todd, 2012; Xu et al., 2016).

We find that an important contributor to the peaks in dustiness in spring and summer is an enhanced dust lifetime during those seasons (Fig. 3). A multi-linear regression analysis shows that, on average, approximately one quarter of the variance in seasonal dust loading is explained by the seasonal variability in lifetime, approximately one third is explained by the seasonal variability in emissions, and the rest of the variance is explained by the correlation between emissions and lifetime. In fact, several source regions do not show a clear peak in emissions during spring and summer (i.e., western and eastern North Africa and Southern Africa), but nonetheless show clear peaks in loading and DAOD that are driven by the increased lifetime in those seasons. This seasonality in the lifetime is likely driven by the stronger convection in spring and summer, lofting emitted dust to greater heights in those seasons when solar insolation is largest. One exception might be the Sahel, for which the lifetime in Fall is longer than in Summer (Fig. 3a), which appears to be due to increased wet deposition (Fig. 4f), presumably due to proximity to the Intertropical Convergence Zone and the West African monsoon in summer (Glaser et al., 2015). Nonetheless, we find that most of the variance in dust lifetime between seasons and source regions is explained by differences in the planetary boundary layer depth ($R^2 = 0.86$; Fig. 3b), which in turn are largely due to differences in the top-of-atmosphere shortwave downwelling flux ($R^2 = 0.86$; Fig. 3c).

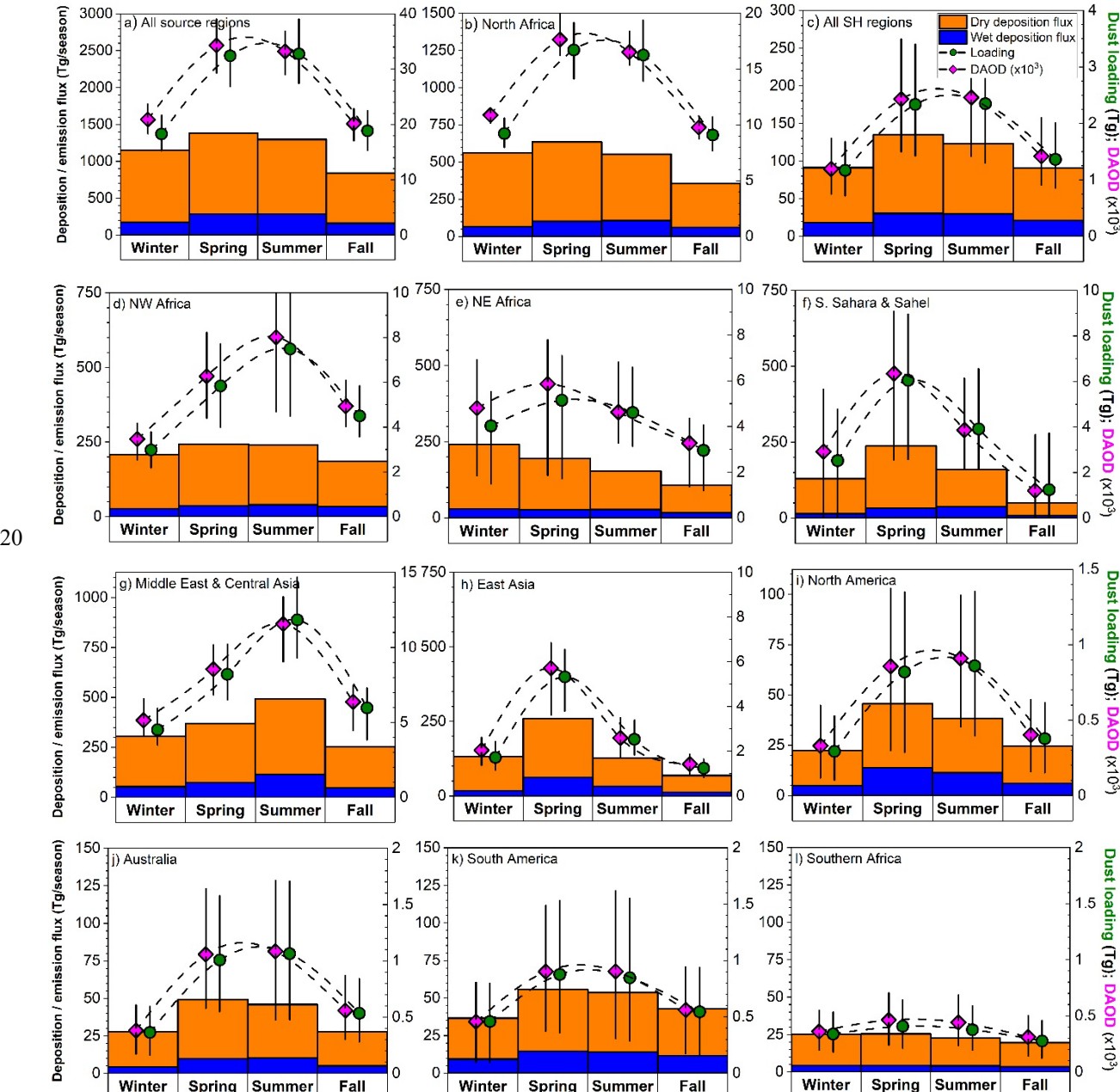

**Figure 4. Seasonal contributions of each source region to the global dust cycle.** Shown are the seasonal cycles of the wet deposition flux (blue bars and left axis), dry deposition flux (brown bars and left axis), dust loading (green circles and right axis), and DAOD (magenta diamonds and right axis) generated by (**a**) all source regions, (**b**) all North African source regions, (**c**) all Southern Hemisphere source regions, and (**d-l**) each of the nine individual source regions. The sum of the seasonal wet and dry deposition fluxes is approximately equal (within a few percent) to the seasonal dust emission flux generated by each source region. Results for loading and DAOD are slightly offset horizontally for clarity. Seasons refer to boreal seasons for global results (panel (**a**)) and for local seasons for all other panels. Note that the

vertical axis scale differs between source regions. Error bars denote one standard error from the median; error bars on deposition fluxes
usually exceeded 100% and are not included for clarity.

### 3.3 Spatial distribution of each source region's contribution to DAOD, column loading, and concentration

Consistent with the ~50% fractional contribution of North African source regions to the global DAOD, we find that North African dust contributes substantially to DAOD and loading in large fractions of the Northern Hemisphere, with dust from East Asia and the Middle East & Central Asia also contributing substantially in several regions (Figs. 5 and S11). However, the intertropical convergence zone (ITCZ) poses a formidable barrier to the inter-hemispheric transport of dust from these major Northern Hemisphere source regions (also see seasonal DAOD and loading results in Figures S12-S19). Even though Northern Hemisphere dust accounts for ~93% of the global dust loading, it contributes only up to ~10% to the dust loading south of the ITCZ. Note that this result depends sensitively on the ability of models to represent interhemispheric transport. Although other modeling studies have found somewhat different contributions of Northern Hemisphere source regions to Southern Hemisphere dust (e.g., Li et al., 2008; Albani et al., 2012), our results support the conclusion that dust in the Southern Hemisphere is overwhelmingly supplied by the Australian, South American, and South African source regions.

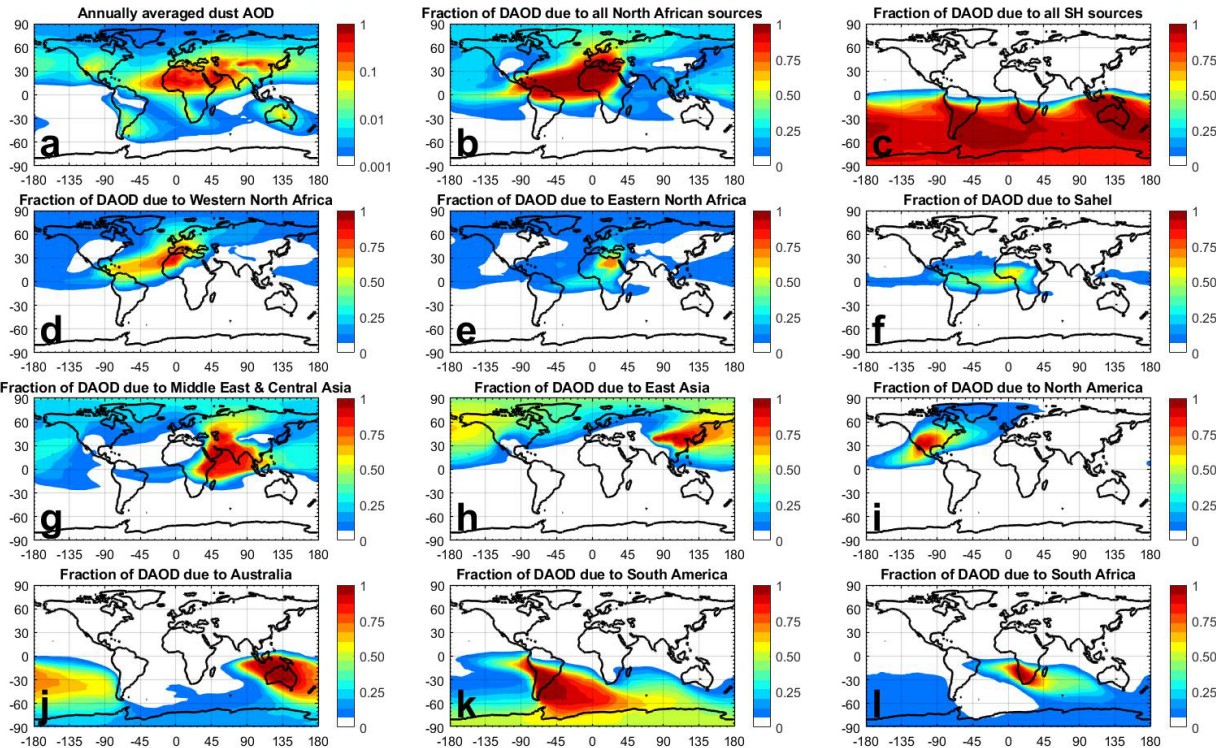

**Figure 5. Attribution of the annually averaged DAOD to the world's main dust source regions.** Shown first is (**a**) the annually averaged DAOD produced from all source regions combined, followed by the fraction of DAOD that is due to (**b**) all North African and (**c**) all Southern Hemisphere source regions. The fraction of DAOD due to each of the three North African source regions are shown in panels (**d**)-(**f**), and

the fraction of DAOD due to the other three Northern Hemisphere source regions of Middle East & Central Asia, East Asia, and North America are shown in panels (**g**)-(**i**). Finally, the fraction of DAOD due to the three Southern Hemisphere source regions of Australia, South America, and South Africa are shown in panels (**j**)-(**l**). Attributions of seasonal DAOD to the different source regions are shown in Figures S12-S15.

Constraints on the zonally averaged dust concentration provide further insight into the contribution of the different source regions throughout the three-dimensional (3D) atmosphere (Fig. 6). We find that the interhemispheric transport of Northern Hemisphere (NH) dust is likely facilitated by strong vertical transport shown by dust emitted from the arid North African and Asian deserts (Figs. 6d-h), which probably plays a role in the longer lifetimes of dust emitted from those regions (Fig. 3). Consequently, the cross-equatorial transport of dust originating from the NH makes the largest fractional contribution to concentration in the SH stratosphere (<200 hPa; Figs. 6d-h), although transport into this region could be distorted by model errors in the middle atmosphere circulation that result from insufficient vertical resolution and an artificially low upper boundary (e.g., Rind et al., 2020). In the SH upper troposphere, dust originates mainly from austral sources (Fig. 6c). Although dust concentrations are small at this altitude (Fig. 6a), dust there could be critical for the heterogeneous nucleation of cirrus (Cziczo et al., 2013) and mixed-phase clouds (Vergara-Temprado et al., 2018), which could have important impacts on climate (Storelvmo, 2017). Overall, we find that ~0.4% of $PM_{20}$ dust emitted in the NH is transported to the SH, whereas only ~0.2% of $PM_{20}$ dust emitted in the SH is transported to the NH. As such, interhemispheric dust transport is quite rare, and transport of dust from the SH to the NH is even less efficient than from the NH to the SH (Figs. 6c, j, k, l). Likely reasons for this include that SH dust has a lower average lifetime than NH dust (Fig. 3) and is emitted further from the equator and the summer ITCZ than dust from for instance North Africa and the Middle East.

It is important to note that the inverse model's vertical distribution of dust is largely determined by that simulated by the models in our ensemble as we have not incorporated observational constraints on dust vertical profiles. Since comparisons of model simulations against data from the Cloud-Aerosol Lidar with Orthogonal Polarization (CALIOP) and the Cloud–Aerosol Transport System (CATS) airborne lidars indicate substantial discrepancies (Yu et al., 2010; Kim et al., 2014; Koffi et al., 2016; O'Sullivan et al., 2020b), results presented here likely suffer from similar biases. A future version of this product could thus use dust extinction profiles retrieved from CALIOP (Omar et al., 2010; Song et al., 2021), CATS (Yorks et al., 2014), or in situ measurements (Brock et al., 2019) to constrain the vertical distribution of dust.

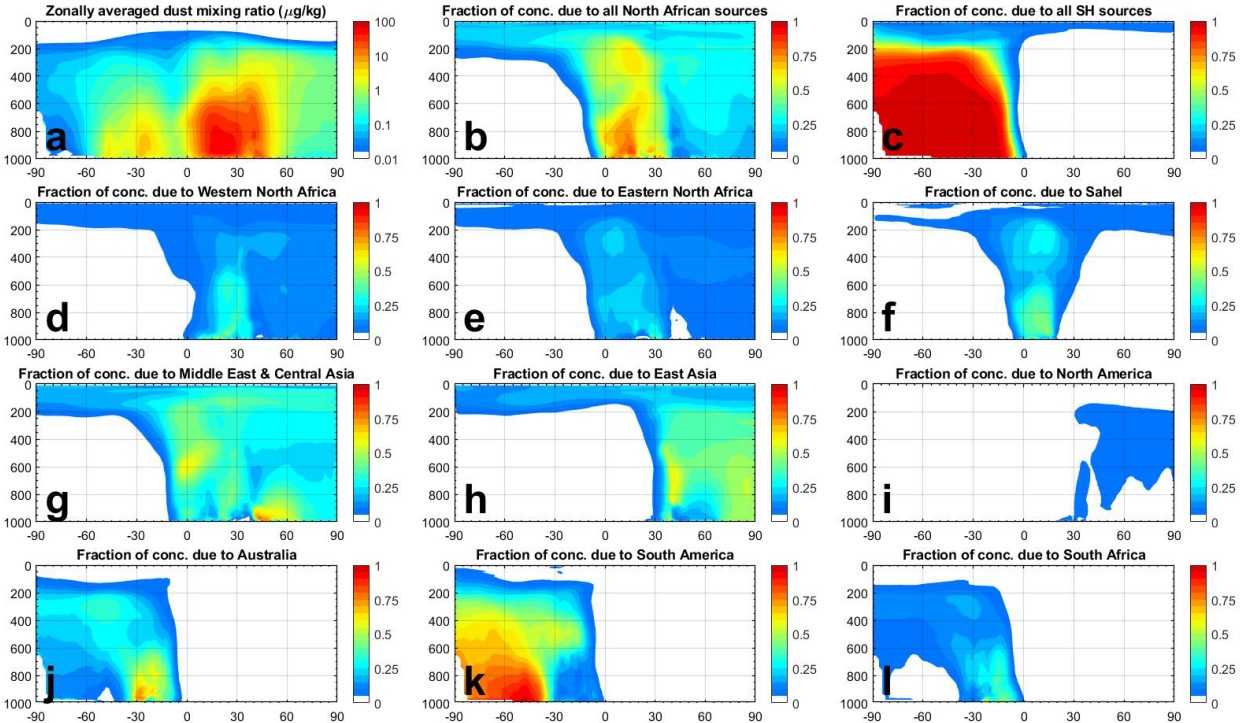

**Figure 6. Attribution of the zonally averaged and annually averaged PM$_{20}$ dust concentration to the world's main source regions.** Panel (a) shows the zonally averaged dust mixing ratio (dust concentration normalized by air density) as a function of latitude (horizontal axis) and pressure in hPa (vertical axis). Panels (b)-(l) show the partition of the dust concentration per source region, with panel ordering identical to Figure 5. The seasonally resolved attribution of the zonally averaged dust concentration is shown in Figures S20-S23.

## 3.4 Constraints on each source region's contribution to dust deposition fluxes

We used our results to attribute the dust deposition flux to the different major source regions (Figs. 7 and 8a). These results strongly mirror the attribution of dust loading and DAOD. For instance, North African dust accounts for ~20-100% of dust deposition in much of the Northern Hemisphere (Fig. 7b). Dust from East Asia and the Middle East & Central Asia account for the bulk of the remaining dust deposition and dominate in large regions near their respective source regions (Figs. 7g, 7h, 8a). We also find that Northern Hemisphere dust contributes only a few percent of dust deposition fluxes throughout most of the Southern Hemisphere (Fig. 7c; Tables 2 and 3).

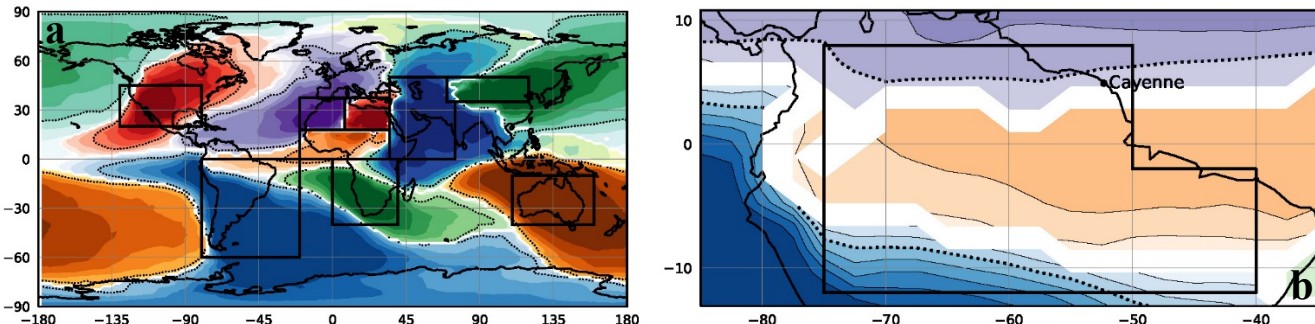


**Figure 7. Attribution of the annually averaged deposition flux of PM_{20} dust to the world's main source regions.** Panel ordering is identical to Figure 5 and the seasonally resolved attribution of the dust deposition flux is shown in Figures S25-28.

**Figure 8. Percentage of dust deposition supplied by the dominant source region at each location for (a) the entire globe and (b) the Amazon rainforest.** Different colors represent different dominant source regions, with shading decreasing in 10% increments from a maximum of 100% to a minimum of 20%. The 50% contour is identified by a black dotted line and white shading denotes areas where two or more dust source regions contribute similarly to the deposition flux. The black boxes in panel (a) denote the nine major source regions, and the black box in panel (b) denotes the boundaries of the Amazon rainforest

used here, based on Yu et al. (2015). Also shown in (b) is Cayenne, the location of the field site where Prospero et al. (2020) obtained dust measurements (see text).

### 3.4.1 Dust deposition to high albedo regions

We also constrained the deposition fluxes to land regions for which dust deposition produces important impacts (Table 2).
This includes the snow and ice-covered regions of Antarctica, the Arctic, and the Tibetan plateau (Lee et al., 2017; Kylling et al., 2018; Sarangi et al., 2020). Dust deposition to these high albedo regions can darken snow and icepacks, thereby producing warming and accelerating melting (Painter et al., 2010; Shao et al., 2011; Mahowald et al., 2014; Skiles et al., 2018). We find that the Tibetan plateau, being adjacent to major East Asian source regions (e.g., the Taklimakan Desert), receives a large deposition flux of 16 (11-25) Tg/year. This is consistent with findings that snowpack darkening from increasing dust deposition
has played an important role in the regional warming of this region (Lau et al., 2010; Sarangi et al., 2020).

Greenland receives a relatively smaller dust deposition flux of 0.19 (0.07-0.52) Tg that is supplied by several different source regions, namely Northern Africa (30 (19-45) %), East Asia (26 (14-32) %), and North America (26 (11-50) %). (Note that we constrain deposition of desert dust only, and thus do not include deposition of dust from high latitude sources supplied by
glacial sediments, which is likely an important contributor at the coastal margins of Greenland (Bory et al., 2003; Bullard et al., 2016).) Our finding that North American dust contributes substantially to dust deposited to Greenland (Figs. 7i, 8a) seems to disagree with geochemical data, which thus far has not identified a clear contribution from North American dust (Bory et al., 2003). This suggests that our results might overestimate North American dust emission, which is approximately a factor of five larger than predicted by most models in both the AeroCom phase I ensemble and in our own model ensemble (Fig. 2
and Table 1). This overestimation would most likely be due to an overestimate of the DAOD over North America that we used to constrain regional emission (Kok et al., 2021). This regional DAOD was obtained from an ensemble of aerosol reanalysis products (Adebiyi et al., 2020), which might thus overestimate DAOD over North America. Furthermore, the inverse model finding of roughly equal contributions of the East Asian and North African source regions to dust deposition to Greenland is in mixed agreement with geochemical data. These data indicate that East Asia is the main source of dust deposited to several
ice core sites in the interior of Greenland (Bory et al., 2002; Bory et al., 2003), with dust from North Africa probably contributing as a secondary source (Lupker et al., 2010). This possible disagreement between the inverse model results and geochemical data is noteworthy, as the inverse model shows a substantially greater contribution from East Asian dust and a smaller contribution from North African dust than most models in both model ensembles (see Fig. 2 and Table 1). As such, the underestimation of Asian dust and overestimation of African dust relative to results from geochemical studies would be
larger for models in the two ensembles than for the inverse model. This finding suggests that current models either substantially underestimate transport of East Asian dust or overestimate transport of North African dust to Greenland.

**Table 2. Constraints on the contribution of dust deposition from each source region to land areas where dust deposition produces important impacts, namely the high albedo regions of Antarctica, Greenland, and the Tibetan Plateau, and the Amazon rainforest.**

Listed are median values, with one standard error intervals listed in parentheses. The coordinates of the Amazon rainforest follow the definition given in Yu et al. (2015), and the Tibetan plateau was taken as the region with elevation over 4000m between 26-40N and 75-105E, based on Easter et al. (2004). These regions are plotted in Fig. S24.

| Source region: | Antarctica | Greenland | Tibetan plateau | Amazon rainforest | All land regions |
|---|---|---|---|---|---|
| Total deposition from all source regions (Tg/year) | 0.14 (0.03-0.55) | 0.19 (0.07-0.52) | 16 (11-25) | 8.5 (2.9-9.7) | $3.8\ (2.7\text{-}7.5) \times 10^3$ |
| Percentage from: | | | | | |
| North African source regions | 0 (0-1) | 30 (19-45) | 4 (2-6) | 90 (86-94) | 47 (45-50) |
| Southern Hemisphere source regions | 99 (98-100) | 0 (0-5) | 0 (0-0) | 6 (4-12) | 7 (5-8) |
| Western North Africa | 0 (0-0) | 18 (13-32) | 1 (1-2) | 41 (28-53) | 19 (14-22) |
| Eastern North Africa | 0 (0-0) | 6 (4-12) | 2 (0-3) | 11 (6-24) | 17 (7-22) |
| S. Sahara & Sahel | 0 (0-1) | 3 (1-5) | 1 (0-1) | 36 (16-53) | 13 (5-22) |
| Middle East & Central Asia | 0 (0-1) | 11 (7-19) | 18 (14-30) | 2 (2-3) | 29 (26-32) |
| East Asia | 0 (0-0) | 26 (14-32) | 77 (67-82) | 0 (0-0) | 14 (11-17) |
| North America | 0 (0-0) | 26 (11-50) | 0 (0-0) | 0 (0-0) | 3 (1-4) |
| Australia | 18 (6-36) | 0 (0-1) | 0 (0-0) | 0 (0-0) | 3 (2-4) |
| South America | 71 (50-90) | 0 (0-2) | 0 (0-0) | 5 (2-11) | 2 (1-3) |
| South Africa | 8 (3-13) | 0 (0-2) | 0 (0-0) | 1 (0-2) | 1 (1-2) |

We also quantified each source region's contribution to dust deposited to Antarctica. We find that the total dust deposition flux to Antarctica equals 0.14 (0.03-0.55) Tg, of which South America provides the bulk (~70%) of the deposited dust, with smaller contributions from Australia (~20%) and South Africa (~10%), and almost no contribution from NH dust (Fig. 7 and Table 2). As such, South America is the dominant dust source region for almost the entire continent of Antarctica (Fig. 8a). These findings are in good agreement with ice core records, for which geochemical fingerprinting has indicated that most present-day deposited dust originates from South America, with a smaller contribution from Australia (Mosley-Thompson et al., 1990; Souney et al., 2002; McConnell et al., 2007; Delmonte et al., 2008; Bory et al., 2010; Delmonte et al., 2019).

### 3.4.2 Dust deposition to the Amazon rainforest

We also obtained the dust deposition flux to the Amazon rainforest, for which the productivity on timescales of decades to millennia is partially controlled by delivery of phosphorus by settling dust (Swap et al., 1992; Okin et al., 2004). We find that the Amazon rainforest receives an average annual dust flux of 0.9 (0.3-1.1) g m$^{-2}$ year$^{-1}$, which corresponds to a total annual dust deposition flux of 8.5 (2.9-9.7) Tg/year (Table 3). This is similar to results from the (unmodified) simulations in our ensemble, which predict a median Amazon deposition flux of 11.0 Tg and a range of 4.4 to 14.8 Tg. Further, our results are quantitatively similar to the Amazon dust deposition flux of 8 – 10 Tg/year obtained by Prospero et al. (2020) based on dust measurement at Cayenne (French Guiana) and the dust product of the Modern-Era Retrospective analysis for Research and Applications, Version 2 (Buchard et al., 2017; Randles et al., 2017). Note that these results might somewhat underestimate deposition fluxes because most current models are unable to simulate the long-range transport of super-coarse dust with $D >$ 10 μm (Ansmann et al., 2017; Weinzierl et al., 2017).

The Amazon deposition fluxes found here (for the years 2004-2008) and in Prospero et al. (2020) are a factor of ~2-3 less than the 28 (8-50) Tg/year obtained from an analysis of 2007-2013 data from the CALIOP satellite instrument by Yu et al. (2015). Our lower estimate of dust deposition for the Amazon rainforest is expected because the CALIOP study substantially underestimated the dust extinction efficiency far from source regions. Indeed, the CALIOP estimates used a mass extinction efficiency (MEE) of 0.37 m$^2$/g after Kaufman et al. (2005), which is less than the globally integrated mass extinction efficiencies of ~0.68 and ~0.54 m$^2$/g predicted by AeroCom models and obtained here (Table 1), respectively. Furthermore, the MEE for dust near the Amazon rainforest is larger than the globally integrated MEE (Figure S10) because most coarse particles deposit during long-range transport across the Atlantic. Indeed, the recent DustCOMM dataset (Adebiyi et al., 2020), which explicitly accounts for the enhancement of dust extinction due to particle asphericity, shows an MEE of ~0.8 – 1.0 m$^2$/g after trans-Atlantic transport of North African dust. This is in good agreement with available measurements (Li et al., 1996; Denjean et al., 2016; Figs. 8 and 9 in Adebiyi et al., 2020). We similarly find that the MEE over the Amazon rainforest is 0.86 (0.76-1.05) m$^2$/g, such that the CALIOP-derived results likely underestimate the MEE by a factor of ~2-3, thereby overestimating deposition fluxes by the same factor. Taking this bias in the assumed MEE into account would bring the CALIOP-derived deposition fluxes in agreement with our results. These results emphasize the need for accurate and spatially-varying constraints on the MEE, such as provided here and in Adebiyi et al. (2020) as part of the DustCOMM dataset. Overall, our results add to a growing consensus that, although the dust-borne delivery of phosphorus is likely critical for the long-term productivity of the Amazon rainforest, these fluxes are substantially less than previously thought and are rivaled by the delivery of more soluble phosphorus by biomass burning aerosols from Southern Africa (Barkley et al., 2019; Prospero et al., 2020).

Our finding of a substantially lower deposition flux to the Amazon rainforest illustrates the advantages of integrating observational, experimental, and modeling constraints. Results from analyses of model simulations and satellite data are

subject to possibly substantial biases due to a number of required assumptions, including regarding the optical properties and size distribution of dust. In our approach, dust properties are instead based on observational constraints for which the uncertainties have been propagated into our results (Kok et al., 2017; Adebiyi and Kok, 2020), and for which quantitative predictions can be evaluated against independent measurements (see Kok et al., 2021). In addition, our approach integrates a variety of regional measurements ranging from AOD to dust size distributions along with model estimates of transport, in contrast to observational estimates of nutrient supply based upon a more limited range of observations or retrievals. Two further advantages of the constraints presented here are that they are source region-resolved and are available globally. This former factor is particularly important for accounting for the effects of regional differences in soil mineralogy on dust impacts on radiation (Perlwitz et al., 2015a; Scanza et al., 2015), clouds (Liu et al., 2012; Atkinson et al., 2013; Shi and Liu, 2019), and biogeochemistry (Zhang et al., 2015), as well as for interpreting records of dust deposition from natural archives (Albani et al., 2015).

Our results also provide insight into the source regions that provide dust to the Amazon rainforest (Fig. 8b). Previous studies have argued that either the Bodélé depression (Koren et al., 2006) or El Djouf (Yu et al., 2020) are dominant contributors to dust deposition to the Amazon. However, we find that the western North Africa source region (containing El Djouf) and the Southern Sahara & Sahel source region (containing the Bodélé depression) contribute roughly equally to the Amazonian dust deposition flux. Indeed, western North Africa contributes 41 (28-53) % and dominates the deposition flux in the northern part of the Amazon rainforest, whereas the Southern Sahara & Sahel contributes 36 (16-53) % and dominates in the central and Eastern parts of the Amazon rainforest. The contribution of both the western North Africa and Southern Sahara & Sahel source regions peak in boreal Spring, with smaller contributions in Winter and Summer (Figs. S25-S28). Most of the remainder of the dust deposition flux is supplied by eastern North African dust (11 (6-24) %) and notably by South American dust (5 (2-11) %), which dominates in the southwestern part of the Amazon. Overall, our results thus indicate that a large number of different source regions make important contributions to dust deposition to the Amazon rainforest.

### 3.4.3 Dust deposition to oceanic regions

We further used our results to quantitatively constrain the $PM_{20}$ dust deposition flux of each source region to each ocean basin (Table 3). Dust deposition provides critical nutrients, such as phosphorus and iron, to open ocean regions such as the Southern Ocean and the North Pacific where primary productivity can be limited by the supply of these dust-borne nutrients (Jickells et al., 2005; Myriokefalitakis et al., 2018). Consequently, glacial-interglacial variations in atmospheric dust deposition flux to the oceans are hypothesized to have modulated atmospheric $CO_2$ concentrations (Martin, 1990; Ridgwell and Watson, 2002; Lambert et al., 2008). We constrain the total flux of $PM_{20}$ dust deposited to oceans to be 0.8 (0.6-1.3) $\times$ $10^3$ Tg. This is approximately double the median deposition flux to oceans that was obtained in recent model ensemble studies (Table 5 in Checa-Garcia et al., 2020; Tables 8 and 9 in Wu et al., 2020). This larger dust deposition flux to oceanic regions likely occurs because we correct the underestimation (or omission) by current models of dust with geometric diameters between 10 to 20

μm (Adebiyi and Kok, 2020), which we find makes up approximately half (46 (35-66) %) of this deposition flux to oceans. Although the bulk of this dust deposition to oceans occurs in oceanic basins surrounding major source regions (Fig. 7a), especially for coarse dust, we also find significant fluxes into more remote nutrient-limited ocean basins, such as a deposition flux of 25 (6-51) Tg/year to the Southern Ocean (Table 3), much of which originates from South America.

**Table 3. Constraints on the contribution of dust deposition from each source region to the world's main ocean basins.** Listed are median values, with one standard error intervals listed in parentheses. Coordinates of ocean basins are based on NOAA's World Ocean Atlas (Locarnini et al., 2010). All regions are plotted in Fig. S24 for clarity.

| Source region: | South Atlantic Ocean | North Atlantic Ocean | South Pacific Ocean | North Pacific Ocean | Indian Ocean | Mediterra nean Sea | Southern Ocean | Arctic Ocean | All ocean regions |
|---|---|---|---|---|---|---|---|---|---|
| Total deposition from all source regions (Tg/year) | 86 (51-149) | 229 (168-344) | 19 (10-34) | 74 (50-106) | 189 (121-314) | 57 (35-78) | 25 (6-51) | 2 (1-6) | 0.8 (0.6-1.3) × 10³ |
| Percentage from: | | | | | | | | | |
| North African source regions | 4 (2-8) | 95 (91-98) | 1 (0-2) | 8 (4-15) | 5 (2-6) | 99 (96-99) | 0 (0-0) | 28 (22-51) | 38 (33-43) |
| Southern Hemisphere source regions | 95 (91-97) | 0 (0-1) | 98 (97-100) | 0 (0-2) | 12 (5-17) | 0 (0-1) | 100 (99-100) | 0 (0-1) | 21 (15-27) |
| Western North Africa | 0 (0-1) | 58 (43-73) | 0 (0-1) | 3 (2-6) | 1 (0-1) | 36 (20-42) | 0 (0-0) | 17 (13-37) | 21 (15-26) |
| Eastern North Africa | 1 (0-4) | 8 (2-17) | 0 (0-1) | 3 (1-6) | 3 (1-4) | 62 (55-75) | 0 (0-0) | 7 (4-13) | 10 (3-13) |
| S. Sahara & Sahel | 2 (1-5) | 22 (9-44) | 0 (0-1) | 1 (1-3) | 1 (0-1) | 2 (1-3) | 0 (0-0) | 2 (1-4) | 7 (3-15) |
| Middle East & Central Asia | 1 (0-1) | 2 (1-3) | 0 (0-1) | 10 (7-22) | 83 (78-90) | 1 (1-3) | 0 (0-0) | 23 (7-35) | 31 (27-38) |
| East Asia | 0 (0-0) | 1 (0-1) | 0 (0-0) | 55 (41-76) | 0 (0-1) | 0 (0-0) | 0 (0-0) | 27 (21-33) | 5 (3-9) |
| North America | 0 (0-0) | 2 (0-5) | 0 (0-0) | 14 (7-33) | 0 (0-0) | 0 (0-0) | 0 (0-0) | 15 (5-33) | 3 (2-4) |
| Australia | 0 (0-1) | 0 (0-0) | 68 (20-81) | 0 (0-0) | 7 (3-11) | 0 (0-0) | 10 (4-30) | 0 (0-0) | 3 (1-5) |
| South America | 68 (28-84) | 0 (0-1) | 28 (15-78) | 0 (0-1) | 1 (0-2) | 0 (0-0) | 88 (63-94) | 0 (0-0) | 12 (6-18) |
| South Africa | 24 (11-65) | 0 (0-0) | 2 (0-3) | 0 (0-1) | 3 (1-7) | 0 (0-0) | 2 (1-6) | 0 (0-1) | 5 (3-7) |

## 4. Discussion

## 4.1 Limitations of the methodology

We consider the results presented in the previous section to be more accurate constraints on the source region-resolved dust loading, concentration, DAOD, and deposition flux than results obtained directly from regional and global model simulations

(e.g., Tanaka and Chiba, 2006; Mahowald, 2007; Huneeus et al., 2011; Wu et al., 2020). Nonetheless, our results are subject to a number of important limitations. Critically, the results obtained here depend on a number of previous products, including the analytical framework to join observational and modeling constraints on the global dust cycle developed in our companion paper (Kok et al., 2021), the ensemble of climate and atmospheric chemistry model simulations, and constraints on the globally averaged dust size distribution (Adebiyi and Kok, 2020) and the regional DAOD (Ridley et al., 2016). The results presented

here are therefore subject to the limitations of these studies, which are discussed in detail in the corresponding papers.

Some limitations from these previous studies are of particular relevance in the interpretation of our results. In particular, Ridley et al. (2016) obtained the regional DAOD in dust-dominated regions by taking the total AOD - which was calculated from satellite-retrieved AOD that was bias-corrected using ground-based aerosol optical depth measurements from the AErosol RObotic NETwork (AERONET) – and subtracting the AOD from non-dust aerosol species simulated by an ensemble of four

model simulations. Although the error quantified by the spread in the model-simulated non-dust AOD for each region was propagated into the results here, a systematic bias in model simulations of non-dust AOD would cause a systematic bias in the regional DAOD in the Ridley et al. (2016) product. Such a bias could arise in particular for regions in which insufficient observations are available to constrain models, and in which biomass burning and anthropogenic aerosols supply a large

fraction of AOD, as models rely on emission inventories to simulate those (Lamarque et al., 2010). Errors in emission inventories would thus affect most or all models and cause a potentially substantial bias in the regional DAOD that is not accounted for in our error analysis. Such a bias could in particular affect our constraints on Asian source regions, for which there are also fewer in situ dust measurements available than for North African dust (e.g., Mahowald et al., 2009; Kok et al., 2021). Furthermore, our constraints on the lesser source regions of North America, Australia, South America, and South Africa

rely on regional DAOD constraints obtained from an ensemble of aerosol reanalysis products (Adebiyi et al., 2020) that assimilate ground- and satellite-based AOD retrievals. These products could similarly be biased because of difficulties distinguishing dust from other aerosol species (see further discussion in Kok et al., 2021). It is thus possible that some of the differences between our results and model simulations, for instance the larger contribution of North American and Asian sources found here than in model ensembles, could be due to such a bias in our results, model results, or both.

An additional limitation arises from possible systematic biases in dust transport simulated by the ensemble of models. If models in our ensemble have consistent biases in representing dust transport from different source regions, then this would cause dust from one source region to erroneously be assigned to another source region. For instance, if models systematically underestimate the long-range transport of North African dust across East Asia and the North Pacific (Hsu et al., 2012b), then

this could cause our results to overestimate the contribution of East Asian dust. Such an underestimation of long-range transport could be due to a systematic overestimation of dust deposition, as indicated by some studies (Ridley et al., 2012; Yu et al., 2019).

Finally, the climatology of the contributions of different source regions to the global dust cycle obtained here is based mostly on model data and regional DAOD constraints for the period 2004-2008 (Ridley et al., 2016; Kok et al., 2021). As such, differences in the relative contributions of the main source regions before or after that period are not reflected in our results. This includes an increase in dust loading in the Middle East (Hsu et al., 2012a; Kumar et al., 2019) and a decrease in dust loading in East Asia (Shao et al., 2013; Shimizu et al., 2017; An et al., 2018; Tai et al., 2021).

Although our methodology accounts for and propagates a large number of (quantifiable) errors in our analysis (Kok et al., 2021), the limitations above raise the possibility that our analysis nonetheless underestimates the true errors on the contribution of different source regions to the global dust cycle. We found in our companion paper that errors on results summed over all source regions are likely realistic for the NH because the inverse model reproduced independent measurements of surface concentration and deposition flux (well) within the combined uncertainties of the experimental data and the inverse model (see Figs. 7 and 10 in Kok et al., 2021). For the SH, errors were only slightly smaller than needed to reproduce measurements within the uncertainties. However, these findings do not necessarily mean that errors on the relative contribution per source region are realistic. For instance, the limitations discussed above could produce biases that differ between source regions and that may partially cancel out in comparisons against measurements of surface concentration and deposition flux.

To inform whether our errors on the contribution to the global dust cycle per source region are realistic, we calculated the optimal contribution of each source region that minimizes errors with respect to compilations of surface concentration and dust deposition flux measurements. This procedure is detailed in the Supplementary Methods and is similar to using DAOD constraints in the inverse model (Section 2.3 in Kok et al., 2021). We expect the optimization using surface concentration and deposition flux to yield a larger range of possible values (i.e., larger errors) than the optimization using DAOD constraints because model simulations of surface concentration and deposition flux are subject to greater model errors. This is because these results depend to a larger extent on correctly simulating deposition and the dust vertical profile, which models struggle with (Huneeus et al., 2011; Yu et al., 2019; O'Sullivan et al., 2020a). Furthermore, dust deposition flux measurements are subject to large experimental and representation errors (see Section 3.1 in Kok et al., 2021), such that these measurements are less potent in constraining simulations of the global dust cycle than DAOD constraints. This is shown by the greater than half-an-order-of-magnitude error of comparisons of inverse model and individual simulation results against deposition flux measurements (Figs. 7, 10, S10, and S13 in Kok et al., 2021).

We find that the optimization against surface concentration data yields median results consistent with those of the inverse model's optimization against DAOD constraints for both NH and SH source regions, but with a larger 95% confidence interval, as expected (Fig. 9). The optimization against deposition flux measurements does show differences from the inverse model results using DAOD constraints, with on average a larger fractional contribution of Asian source regions and a smaller fractional contribution of North African source regions, thereby reinforcing our finding that some models overestimate North

African dust and underestimate Asian dust. In the SH, we find that larger contributions from Australian and South American dust are needed to match deposition flux measurements, particularly in Antarctica (see Figure 10 in Kok et al. (2021)). Overall, these comparisons do not show that errors on the fractional contributions of the Northern Hemisphere are underestimated, though we nonetheless interpret these errors as lower bounds in light of the limitations discussed above. Conversely, errors on the contributions of the Southern Hemisphere source regions are likely underestimated, presumably for the reasons reviewed above.

An important finding in Fig. 9 is that using deposition flux measurements to constrain models yields substantially different results than using DAOD and surface concentration data. This likely occurs in part because deposition flux measurements are subject to much greater experimental and representation errors and in part because model errors in deposition are substantially larger than model errors in DAOD and surface concentration (Cakmur et al., 2006; Huneeus et al., 2011; Albani et al., 2014; Stanelle et al., 2014; Kok et al., 2021). In turn, these larger model errors likely occur in part because of substantial errors in parameterizations of dry and wet deposition (van der Does et al., 2018; Yu et al., 2019; Emerson et al., 2020), and in part because deposition occurs at the end of the dust lifecycle, such that model simulations of deposition are also affected by model errors in simulating dust emission and horizontal and vertical transport. Consequently, using deposition flux data sets to constrain the modern-day dust cycle could result in large errors, whereas using surface concentration and DAOD data likely results in more accurate constraints on the dust cycle. Reconstructions of dust in the Last Glacial Maximum and other previous climates usually rely only on deposition flux data sets and are thus more susceptible to these substantial errors in measurements and model simulations of dust deposition (e.g., Albani et al., 2016; Ohgaito et al., 2018).

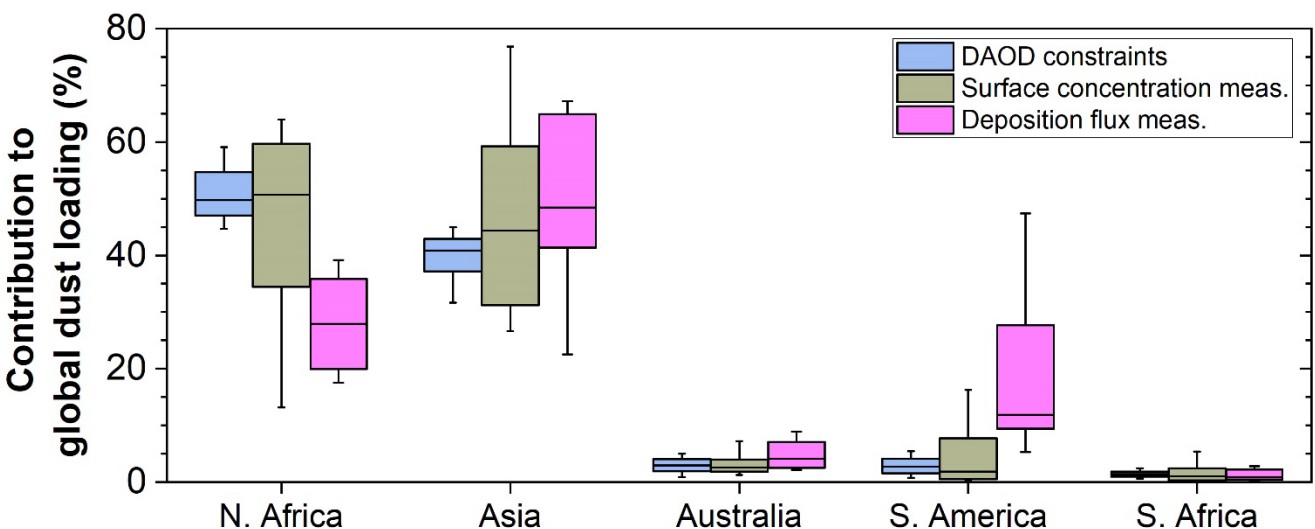

**Figure 9.** Optimal fractional loading per source region that optimizes agreement against constraints on DAOD (blue bars; these are the inverse model results reported in this paper and in Kok et al. (2021)) and against measurement compilations of surface concentration (dark

yellow bars) and deposition flux (pink bars). Box boundaries and whiskers respectively denote the 68% and 95% probability range and horizontal solid lines denote the median result. The North American source region was omitted, and the three North African source regions (western North Africa, eastern North Africa, and Southern Sahara & Sahel) and the two Asian source regions (Middle East & Central Asia and East Asia) were grouped together to avoid overfitting (see Supplementary Methods for further details).


Note that (ensembles of) model simulations are affected by biases similar to those described above. Indeed, although our errors likely underestimate the true errors for at least the SH source regions, our methodology mitigates model errors by drawing from an ensemble and by using constraints on dust abundance and dust microphysical properties to correct model biases, resulting in substantially improved agreement against independent data (Kok et al., 2021). As such, our approach here, though

subject to important limitations and biases, is likely more accurate than (ensembles of) model simulations. Overall, these issues underscore the need to better constrain model simulations and aerosol reanalysis products by obtaining more measurements of dust and other aerosol species outside of the regions affected by North African dust and by obtaining more accurate constraints on speciated AOD (Kahn and Gaitley, 2015).

### 4.2 Implications for improving global dust cycle models and constraining dust impacts on the Earth system

Our constraints on each dust source region's size-resolved contribution to the global dust cycle can be used to constrain a variety of important dust impacts on the Earth system. For instance, combining our results with spatially resolved and particle-size resolved soil-surface mineralogy (Claquin et al., 1999; Journet et al., 2014) can help constrain the regionally-varying mineralogy of dust aerosols (Kok, 2011; Perlwitz et al., 2015a; Scanza et al., 2015; Pérez Garcia-Pando et al., 2016; Li et al., 2020). This has the potential to advance our understanding of a variety of important dust impacts that depend on mineralogy,

including dust impacts on the radiation budget (Scanza et al., 2015), on cirrus and mixed-phase clouds (Atkinson et al., 2013), on atmospheric chemistry (Cwiertny et al., 2008), and on biogeochemical cycles (Mahowald, 2011; Ito et al., 2019; Hamilton et al., 2020). Of particular note in this regard is the synergy with the data on surface mineralogy of dust source regions, such as from the upcoming launch of the NASA Earth Surface Mineral Dust Source Investigation (EMIT) mission (Green et al., 2020).


A second important implication of our results is that our data can be used to inform the interpretation of dust deposition records from natural archives in the modern climate (Mahowald et al., 2010; Albani et al., 2012; Delmonte et al., 2013; Hooper and Marx, 2018). In particular, the constraints on the relative contribution of each source region to the deposition flux at measurement sites could inform the interpretation of changes in dust deposition fluxes through time.


Our results raise the question of why models produce a biased ranking of dust sources that need to be corrected by observationally informed constraints. Part of the cause of model biases is likely the limited knowledge of the emission environment. For instance, models may not correctly characterize the effects of topography, roughness, and vegetation that

partially absorb the force of the wind and shield soil particles from erosion (King et al., 2005; Okin, 2008; Menut et al., 2013;

Ito and Kok, 2017). Models also are limited by inadequate information on soil properties, such as soil size distribution and aggregation state, and how those vary in space and time (Shao, 2001; Kok et al., 2014b). In addition, the relation between wind speed on the scale of the grid box and dust source is difficult to parameterize (Lunt and Valdes, 2002; Cakmur et al., 2004; Ridley et al., 2013; Comola et al., 2019), yet has a substantial effect because dust emissions are threshold dependent and scale non-linearly with wind speed (Shao, 2008; Kok et al., 2012). Furthermore, the mismatch between the wind speed at the resolved

grid-box scale and that at the local dust source scale will vary regionally, partly as a result of sub-grid variations in topography. For instance, the gap between the Tibesti and Ennedi mountains upwind of the Bodélé Depression is below the resolution of many global dust models but enhances the wind over the dust source compared to a grid-box average: a strengthening that is endemic and does not extend to most other grid boxes. Furthermore, models do not correctly capture some of the mesoscale meteorological events that might play an important role in the dust cycle (Schepanski et al., 2009), including haboobs (Pantillon

et al., 2015) and possibly dust devils (Koch and Renno, 2005; Jemmett-Smith et al., 2015; Klose and Shao, 2016). Finally, one other key reason for the difference between model simulations and the contributions of different source regions to the global dust cycle obtained here could be that models simulate mostly natural dust and commonly omit or underestimate anthropogenic dust such as from agricultural sources and fugitive dust, whereas the results obtained here inherently account for mineral dust from all natural and anthropogenic sources (see Kok et al., 2021). As such, differences between our results and model results

could also be due to the contribution of anthropogenic (agricultural and fugitive) dust, which could be substantial (Tegen et al., 2004; Ginoux et al., 2012).

The results obtained here can address these weaknesses of models in accurately simulating emissions per source region by allowing models to tune each source region's dust cycle to match our observationally informed constraints. One approach is

for models to scale their emissions such that a multi-year simulation, ideally over the period 2004-2008 to match the regional DAOD constraints used in our inverse model (Ridley et al., 2016), matches our constraints on the size-resolved yearly emission flux per source region. A preferable approach would be to scale model emissions to match either the loading or DAOD per source region to our constraints. This would be more difficult as the emission flux to generate a certain loading or DAOD depends on the model's simulated dust lifetimes, but this approach would be more accurate as it ties directly to our main

observational constraint, namely on regional DAOD. Although these approaches could substantially improve simulations of the present-day global dust cycle, it would be preferable to improve model physics such that such tuning is no longer necessary. This is particularly important for model predictions of interannual variability and of the global dust cycle for climates other than the present day. For instance, simulations of future changes in the global dust cycle diverge widely (Stanelle et al., 2014; Kok et al., 2018) and might be substantially biased (Evan et al., 2016). In this context, the accuracy of model upgrades to

improve the accuracy of global dust cycle simulations could be verified through comparison against the observationally informed constraints on dust emissions and loading per source region obtained here.

To facilitate the use of this paper's results in improving model simulations and constraining dust impacts on the Earth system, we report the emission flux, loading, DAOD, concentration, and (dry and wet) deposition fluxes that are generated by each of the nine source regions, resolved by location, season, and particle size. These data are publicly available as part of the DustCOMM data set (Adebiyi and Kok, 2020; Adebiyi et al., 2020; Kok et al., 2021), which is available at https://dustcomm.atmos.ucla.edu/. These data include uncertainties, which further increases its potential for improving constraints on the various dust impacts on the Earth system.

## 5. Conclusions

We have constrained the contribution of the world's main dust source regions to the global cycle of desert dust. We did so by building on the improved representation of the global dust cycle that was obtained in the companion paper (Kok et al., 2021). That work used an analytical framework with inverse modeling to integrate observational constraints on the properties and abundance of atmospheric dust with an ensemble of global aerosol model simulations. Here, we analyzed those inverse modeling results to constrain each source region's contribution to particle size-resolved dust loading, concentration, DAOD, and deposition flux.

We find that the global dust loading is partitioned as follows: North African dust contributes ~50%, including ~15% from the Southern Sahara & Sahel; Asian source regions contribute ~ 40%, with three quarters from the Middle East and Central Asia and one quarter from East Asia; and minor source regions contribute ~10%, with one quarter from North America and three quarters from the Southern Hemisphere source regions in Australia, South America, and South Africa (Fig. 2). Models in both the AeroCom Phase I ensemble and in our model ensemble (Kok et al., 2021) on average estimate a larger contribution of North African dust (~65%) and a smaller contribution of Asian dust (~30%) to atmospheric loading. These differences could be due to various factors (see Sections 4.1 and 4.2), including errors in representing dust emission and the relative contributions of different source regions in models, biases in the regional DAOD used in our analytical framework, and the underestimation or omission of anthropogenic dust by models. We further find that Northern Hemisphere dust contributes only a small fraction (<10%) to atmospheric loading and DAOD in the Southern Hemisphere (Figs. 5 and S11), but that this fraction increases with altitude and thus might be important for heterogeneous nucleation of ice crystals in mixed-phase and cirrus clouds (Shi and Liu, 2019). We also find that the deposition flux of dust to the Amazon basin is ~10 Tg, and thereby a factor of 2-3 less than obtained by previous work.

Our results indicate that dust loading in all source regions peaks in local spring or summer (Fig. 4). This is partially due to increased emissions for some source regions, and partially due to an increased dust lifetime in spring and summer for all source regions (Fig. 3a). In turn, this increased lifetime is likely caused by increased convection and thus increased boundary layer depth in the spring and summer seasons (Figs. 3b and c).


The dust source apportionment data sets obtained here are publicly available as part of the DustCOMM data set (Adebiyi et al., 2020; Kok et al., 2021) at http://dustcomm.atmos.ucla.edu. These data include the size-resolved contributions of each source region to dust emission and deposition fluxes, dust column loading, DAOD, and 3D dust concentration. All these gridded data sets are resolved by season and include realistic error bounds propagated from experimental and modeling

uncertainties in the observational constraints on dust properties and abundance. As such, these data can be used to more accurately constrain the various impacts of dust on the Earth system, and particularly those impacts that depend on particle size or the source-dependent dust mineralogy. This includes dust direct radiative effects and dust impacts on clouds, the hydrological cycle, and biogeochemical cycles.

## Data availability

Data is available at https://dustcomm.atmos.ucla.edu/data/K21b/.

## Author contributions

JFK designed the study, analyzed model data and wrote the manuscript. DSH, LL, NMM, and JSW performed CESM/CAM4 simulations; AI performed IMPACT simulations; RLM performed GISS ModelE2 simulations and produced Figure 8; PRC and ARL performed GEOS/GOCART simulations; MK, VO, and CPGP performed MONARCH simulations; and SA, YB and

RCG performed INCA simulations. YH analyzed results from AeroCom Phase 1 models. AAA provided observational DAOD constraints. MC and NMM provided valuable comments on study design. All authors edited and commented on the manuscript.

## Competing interests

The authors declare that they have no conflict of interest.

## Acknowledgements

This work was developed with support from the National Science Foundation (NSF) grants 1552519 and 1856389 awarded to J.F.K, from the University of California President's Postdoctoral Fellowship awarded to A.A.A., from the European Union's Horizon 2020 research and innovation programme under the Marie Skłodowska-Curie grant agreement No 708119 awarded to S.A. and No. 789630 awarded to M.K. R. C.-G. received funding from the European Union's Horizon 2020 research and innovation grant 641816 (CRESCENDO), and from JSPS KAKENHI Grant Number 20H04329 and Integrated Research

Program for Advancing Climate Models (TOUGOU) Grant Number JPMXD0717935715 from the Ministry of Education, Culture, Sports, Science and Technology (MEXT), Japan to A.I.  P.R.C. and A.R.-L. acknowledge support from the NASA

Atmospheric Composition: Modeling and Analysis Program (R. Eckman, program manager) and the NASA Center for Climate Simulation (NCCS) for computational resources, Y.H. acknowledges NASA grant 80NSSC19K1346, awarded under the Future Investigators in NASA Earth and Space Science and Technology (FINESST) program, and R.L.M. acknowledges support from the NASA Modeling, Analysis and Prediction Program (NNG14HH42I). S.A. acknowledges funding from MIUR (Progetto Dipartimenti di Eccellenza 2018-2022). C.P.G.P. acknowledges support by the European Research Council (grant no. 773051, FRAGMENT), the EU H2020 project FORCES (grant no. 821205), the AXA Research Fund, and the Spanish Ministry of Science, Innovation and Universities (RYC-2015-18690 and CGL2017-88911-R). M.K. and C.P.G.P. acknowledge PRACE for awarding access to MareNostrum at Barcelona Supercomputing Center to run MONARCH. L.L. acknowledges support from the NASA EMIT project and the Earth Venture – Instrument program (grant no. E678605). Y.B. and R.C.-G. benefited in this study from funding by the PolEASIA ANR project under the allocation ANR-15-CE04-0005. We also acknowledge high-performance computing support from Cheyenne (doi:10.5065/D6RX99HX) provided by NCAR's Computational and Information Systems Laboratory, sponsored by the National Science Foundation. We further thank Anna Benedictow for assistance in accessing the AeroCom modeling data, the AeroCom modeling groups for making their simulations available.

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
