# Peer review of "Contribution of the world's main dust source regions to the global cycle of desert dust"

_Atmospheric Chemistry and Physics, 2021_

## Referee Comment (RC1)

Review Comments

This study provides a data set that constrains the relative contribution of each of nine major source regions to size resolved dust emission, atmospheric loading, optical depth, concentration, and deposition flux by integrating an ensemble of global model simulations with observational constraints on the properties and abundance of atmospheric dust. The authors show that current models might on average overestimate the contribution of North Africa sources and underestimate the contribution of Asian dust. The manuscript is well written, and results are clearly presented. This study is a valuable contribution to improving global dust cycle in models and constrain dust impacts on the Earth System. I only have minor comments and recommend publication after they have been answered.

General Comments:
(1) The analysis for global dust cycle in AeroCom Phase I simulations by Huneeus et al. (2011) is a benchmark for research on dust modeling and has been widely used to show the large diversities in simulating global dust cycle. Since new features and parameterization schemes have been developed and added to global climate models for the past ten years, how the fast-developing climate models affect the results? Would you expect better agreement with inverse model results for CMIP6 models?

(2) In Figure 2, the model ensemble is better than AeroCom Phase I model results compared with the inverse model results. It is interesting to see that the estimation for Southern Sahara and Sahel from model ensemble is quite close to the inverse model. I wonder if the authors could explain a little more. The AeroCom simulations are for dust cycle in the year 2000. In the companion paper, the models in the ensemble are all nudged to reanalysis during 2004-2008. For each model, dust size range is extended to 20 μm. Many models use the dust emission scheme of Kok et al. (2014) and have dust size distributions consistent with Kok et al. (2017). How much would the models selected in the model ensemble affect the results? I wonder how the authors interpret the differences between the inverse model and model ensemble for dust loading and DAOD. Do representations of dust transport and deposition in the model play a role here?

(3) The authors talked about using dust extinction profiles from CALIOP and CATS to further constrain dust vertical profiles. How about dust concentration measurements from aircraft campaigns, such as ATom? Is the inverse model able to take measurements from aircraft campaigns? What is the limitation arising from the lack of constraints on dust vertical profile?

(4) In Section 4, the authors give a short discussion on the limitation arising from biases in dust transport. It would be nice if the authors could add more discussion on the limitation arising from representations of dust transport and deposition in models.

Specific comments:
(1) It seems that Eqs 5-8 are quite similar to Eqs 1-4, just for particle size bins. I would suggest the authors remove Eqs 1-4 or Eqs 5-8 for simplicity. These equations all contain θ, φ, and P only for showing coordinate. I would suggest the authors show something simpler, such as $\check{f}_{\tau r,s,k} = \check{\tau}_{r,s,k}/\sum_{r=1}^{N_{sreg}} \check{\tau}_{r,s,k}$, which is more friendly to readers.

(2) Line 338 and 340,  Figs. 6d-h instead of Fig. 6d-h.

(3) Line 497, I think the authors are talking about Table 3 instead of Table 2.

---

## Author Comment (AC1)

**Responses to reviewers' comments** Comments in black and responses in blue**

We thank both reviewers for taking the time to read our paper carefully and their constructive and positive comments, which has helped us to improve the paper. Below we include a point-by-point response to the referee comments and describe the corresponding changes we have made to the manuscript.

We believe these and other changes have addressed the reviewer comments and look forward to possible further suggestions and comments from the referees and editor.

**Referee #1**

This study provides a data set that constrains the relative contribution of each of nine major source regions to size resolved dust emission, atmospheric loading, optical depth, concentration, and deposition flux by integrating an ensemble of global model simulations with observational constraints on the properties and abundance of atmospheric dust. The authors show that current models might on average overestimate the contribution of North Africa sources and underestimate the contribution of Asian dust. The manuscript is well written, and results are clearly presented. This study is a valuable contribution to improving global dust cycle in models and constrain dust impacts on the Earth System. I only have minor comments and recommend publication after they have been answered.

**Thank you for your careful reading of the paper and your helpful and positive comments.**

**General Comments:**

(1) The analysis for global dust cycle in AeroCom Phase I simulations by Huneeus et al. (2011) is a benchmark for research on dust modeling and has been widely used to show the large diversities in simulating global dust cycle. Since new features and parameterization schemes have been developed and added to global climate models for the past ten years, how the fast-developing climate models affect the results? Would you expect better agreement with inverse model results for CMIP6 models?

This is a good question. We have added a few sentences to address this to Section 2.2: "Although newer global aerosol model ensembles are available, such as the AeroCom phase III (Gliss et al., 2021) and CMIP5 model ensembles (Wu et al., 2020), only the dust component of AeroCom Phase I models has been analyzed in sufficient detail (Huneeus et al., 2011) for comparison against the results of our study. However, the error of newer model ensembles relative to various measurements appears to be similar to those for AeroCom Phase I models (see further discussion in Kok et al., 2021) and emissions per source region of CMIP5 models are relatively similar to those of the AeroCom Phase I models analyzed here (see Table 4 in Wu et al., 2020)."

(2) In Figure 2, the model ensemble is better than AeroCom Phase I model results compared with the inverse model results. It is interesting to see that the estimation for Southern Sahara and Sahel from model ensemble is quite close to the inverse model. I wonder if the authors could explain a little more. The AeroCom simulations are for dust cycle in the year 2000. In the companion paper, the models in the ensemble are all nudged to reanalysis during 2004-2008. For each model, dust size range is extended to  $20 \ \mu\text{m}$ . Many models use the dust emission scheme of Kok et al. (2014) and have dust size distributions consistent with Kok et al. (2017). How much would the models selected in the model ensemble affect the results? I wonder how the authors interpret the differences between the inverse model and model ensemble for dust loading and DAOD. Do representations of dust transport and deposition in the model play a role here?

These are good questions and we appreciate the helpful observation that the model ensemble used here better matches the inverse model results than the AeroCom ensemble. You're right that this could be either because of a closer match in the simulated time period or because of improvements in the parameterizations of dust processes (or possibly just random variability between models). To address this, we've added the following sentences to Section 3.1:

We find that the model ensemble used here better matches the fractional contribution obtained by the inverse model than does the AeroCom phase I model. This might be because of a closer match in the simulated time period (most simulations in our ensemble are for the 2004-2008 period for which inverse model results were obtained, whereas AeroCom simulations were for the year 2000) or because of improvements in parameterizations of dust emission and other dust processes.

(3) The authors talked about using dust extinction profiles from CALIOP and CATS to further constrain dust vertical profiles. How about dust concentration measurements from aircraft campaigns, such as ATom? Is the inverse model able to take measurements from aircraft campaigns? What is the limitation arising from the lack of constraints on dust vertical profile?

It's a good point that the inverse model could also take in constraints from in situ measurements of the dust vertical profile, such as from ATom, and we now mention this in Section 3.3. The limitations of the lack of constraints on the dust vertical profile relevant to this paper are mostly that the zonally averaged contributions per source region (Fig. 6) have large uncertainty, and we mention this briefly in Section 3.3. Other important implications are for instance larger uncertainties in deposition and the surface concentration, which is more pertinent to results in the comparison paper, and is briefly mentioned there.

(4) In Section 4, the authors give a short discussion on the limitation arising from biases in dust transport. It would be nice if the authors could add more discussion on the limitation arising from representations of dust transport and deposition in models.

We've added some more discussion to this section, pointing out that an overestimation of deposition, as some studies indicate, would produce an underestimation of longrange transport and thereby produce errors in the partitioning of dust emissions between source regions.

**Specific comments:**

(1) It seems that Eqs 5-8 are quite similar to Eqs 1-4, just for particle size bins. I would suggest the authors remove Eqs 1-4 or Eqs 5-8 for simplicity. These equations all contain q, f, and P only for showing coordinate. I would suggest the authors show something simpler, such as  $\check{f}_{\tau_{r,s,k}} = \check{\tau}_{r,s,k} / \sum_{r=1}^{N_{sreg}} \check{\tau}_{r,s,k}$ , which is more friendly to readers.

Reviewer #2 also thought these equations were unnecessarily complicated. To address this comment, we've followed your helpful suggestions and removed the symbols for the coordinates ( $\theta$ ,  $\phi$ , P) and moved the size-resolved equations to the supplement.

(2) Line 338 and 340, Figs. 6d-h instead of Fig. 6d-h.

**Corrected.**

(3) Line 497, I think the authors are talking about Table 3 instead of Table 2.

**Corrected.**

**Referee #2**

The authors present a very interesting and solid study on the contribution of major dust sources to the global dust cycle, with a unique framework that integrates an ensemble of global model simulations with observational constraints. This work is well designed and of broad interest to the dust community. The writing is clear and thorough. The topic of the paper is well suited for ACP. I recommend the manuscript for publication after only minor revisions.

**General comments:**

(1) The authors should give more "background information" related to the inverse model dataset in section 2. For example, the inverse model includes both natural and anthropogenic dust aerosols. It also excludes the high latitude dust emissions. Moreover, is the sum of the emission from the nine sources equal to the global dust emission in the inverse model? I think it would be important for the readers to make these things clear before moving to the results.

**Thank you for pointing out these omissions, which we have added now to Section 2.**

(2) In section 2.2, the contribution of each source region to global dust loading and DAOD from the AeroCom simulations is scaled by the lifetime and MEE of this work. This method assumes that the ratios of Tr to Tglob and  $\in$ r to  $\in$ glob are roughly similar in the AeroCom model and the analysis in this work. But the inverse model and model ensemble have larger size range than the AeroCom models, which is supposed to cause different lifetime and MEE (Table 1). Would this introduce any uncertainty to the AeroCom estimates? If so, how large is the uncertainty?

That is a good question. Because these are dimensionless ratios of the lifetime or MEE of a particular region to its globally averaged value, these ratios are relatively insensitive to changes in the size distribution or modeled size range. However, the reviewer is correct that there is some sensitivity; for instance, the smaller size range accounted for in AeroCom models causes the lifetime of dust to be controlled to a greater extent by wet deposition. Consequently, differences in the balance between wet and dry deposition between regions would have somewhat different effects depending on the size range modeled. We do expect this error to be quite small relative to the model spread, as for instance indicated by the limited differences in the fractional contributions of different source regions to loading and DAOD (which use these estimated MEE and lifetime ratios) versus the fractional contributions to emission (which is obtained directly from the model simulations). We've added two sentences discussing this to Section 2.2.

**Specific comments:**

Page 2, line 62: I would suggest replacing the words "ice nuclei" with "ice nucleating particles (INP)". The old term "ice nuclei" is thought to be misleading by the ice nucleation community (See section 4.1 in Vali et al., 2015).

**Corrected - thank you for pointing this out.**

Page 5, EQ (1)-(8): I find these equation sets quite complicated and not friendly to the reader. I would suggest the authors to add a few descriptions about the equations in text before presenting them.

Reviewer #1 also thought these equations were unnecessarily complicated. To address these comments and make the equations more friendly for the reader, we've moved the equations for the size-resolved fields to the supplement and have removed the symbols for the coordinates ( $\theta$ ,  $\phi$ , P).

Page 6, line 161 and line 169: By "our analysis", do you mean your model ensemble (as stated in line 154) or your inverse model? If it is model ensemble, why not use the inverse model here?

These references are to results from the inverse model, so we've clarified that at those locations.

Page 7, line 176-179: It seems Figure 2 does not have subplots d to i?

**Corrected.**

Page 7, last paragraph: Please double check the numbers in this paragraph. Some of them are slightly different with Table 1. For example, the dust loading from Sahel ranges from 1.5-5.6 Tg in Table 1 but is written as "1.5-5.7 Tg" in the text (line 197).

Thank you for noticing. We've updated all numbers in the paragraph accordingly.

Page 11, line 255-258: The East Asian dust is also known to be lifted by convection and basin-scale mountain-valley circulation. Why does it have smaller lifetime than dust from North African and the Middle East & Central Asia?

That's a good question. Figure 3b indicates that the PBL in East Asian source regions is less deep than in North African source regions, presumably because the East Asian

deserts are located further north and receive a lower SW flux (Fig. 3c). However, a more detailed analysis would be necessary to understand in more detail the differences in lifetimes between source regions, which is beyond the scope of this paper.

Page 11, line 260-261: Why does Figure S5 present results related to model ensembles instead of the inverse model? It seems this paragraph and Figure 3 mainly discuss the dust lifetime of the inverse model.

That indeed seems a little confusing. The inverse model uses the normalized fields for each of the simulations in the ensemble, such that the lifetime of a particular dust size range is determined by the lifetimes obtained in the model ensemble simulations. We've added some text to clarify this in the caption of Figure 3.

Page 13, line 290-291: What kind of correlation analysis is conducted here? How do you quantitively get the contribution of dust lifetime to the seasonal variation? Or in other words, how do you get the number "one third" here?

This was indeed not very clearly worded. We did a multi-linear regression indicated that, on average, ~25% of the variance in a region's seasonal loading is due to the variation in lifetime (so a bit less than the third noted in the original submission due to an error in the code), another ~35% is due to the variation in the emission, and the rest is jointly explained by the two variables (i.e., is convoluted because of correlation between emission and lifetime). We've clarified the wording to make this clearer in the text.

Page 16, line 345-346: I think Figure 6 only shows that less dust from Southern Hemisphere (SH) is transported to the Northern Hemisphere (NH). It does not have direct implication on the transport efficiency. The weaker emission in the SH (Table 1) may also result in the smaller contribution of SH dust to NH. If the authors want to discuss the transport efficiency here, it would be better to normalize the dust concentrations by the dust emission flux from each source.

The reviewer makes an excellent point that we cannot draw quantitative conclusions about interhemispheric transport efficiency from Figure 6 as we indeed would need to account for the much larger emissions in the NH than in the SH. We have therefore directly calculated the fraction of NH dust that is transported to the SH (~0.4 %) and vice versa (~0.2 %) and now report this in the revised paper.

Page 17, Figure 7: The title for subplot g is not fully visible.

Corrected.

Page 22, line 497: I guess it should be Table 3 instead of Table 2.

**Corrected.**

Page 27, line 631: Shi and Liu (2019) only discusses the dust glaciation effect in the mixed-phase clouds. Please refer to other literatures for the cirrus cloud effect (e.g., Liu et al., 2012).

Good point. We've added a citation to (Liu et al., 2012) as well.

**Additional error analysis**

In addition to addressing the reviewer comments, we have also added an analysis of the realism of the errors on the fractional contribution of the different source regions to the global dust cycle. Specifically, we've added a sensitivity test of our results using optimizations against the dust surface concentration measurements and the deposition flux measurements (instead of using the DAOD constraints). The methodology and results for these optimizations are described in the Supplement (section "Realism of errors in the fractional contributions of different source regions to the global dust cycle" and Fig. S29). We've also added a new Figure 9 and several paragraphs to the end of Section 4.1 that present and interpret these results. Important context for interpreting this addition is the error analysis done in the companion paper, which we revised during the review process. Below, we've copied updated versions of the most pertinent figures, Figures 7a-c and 10a-c in Kok et al. (in press).